# Correlation Dimension of Auto-Regressive Large Language Models

**Xin Du**
Waseda University
duxin@aoni.waseda.jp

**Kumiko Tanaka-Ishii**
Waseda University
kumiko@waseda.jp

## Abstract

Large language models (LLMs) have achieved remarkable progress in natural language generation, yet they continue to display puzzling behaviors—such as repetition and incoherence—even when exhibiting low perplexity. This highlights a key limitation of conventional evaluation metrics, which emphasize local prediction accuracy while overlooking long-range structural complexity. We introduce correlation dimension, a fractal-geometric measure of self-similarity, to quantify the epistemological complexity of text as perceived by a language model. This measure captures the hierarchical recurrence structure of language, bridging local and global properties in a unified framework. Through extensive experiments, we show that correlation dimension (1) reveals three distinct phases during pretraining, (2) reflects context-dependent complexity, (3) indicates a model's tendency toward hallucination, and (4) reliably detects multiple forms of degeneration in generated text. The method is computationally efficient, robust to model quantization (down to 4-bit precision), broadly applicable across autoregressive architectures (e.g., Transformer and Mamba), and provides fresh insight into the generative dynamics of LLMs.

## 1 Introduction

Latest advances in large language models (LLMs) have demonstrated sophisticated capabilities, including mathematical reasoning and planning. These abilities suggest that LLMs internally process information through complex, nonlinear, and potentially hierarchical mechanisms [3], despite operating through simple, token-by-token predictions. Understanding precisely how LLMs achieve such macroscopic behaviors from microscopic predictive steps remains an important open question, essential for grasping the full potential and limitations of these models. Although previous research shows that minimizing next-token prediction loss—perplexity—is theoretically powerful [36], it still falls short in fully explaining unexpected model behaviors, such as hallucinations and repetitive, bland outputs, even when perplexity values are low. Thus, new tools for characterizing the complex behavior of LLMs are urgently needed.

Current evaluation approaches broadly fall into two categories. The first includes methods based on local textual properties, such as lexical or syntactic statistics (e.g., $n$-gram frequencies). These metrics are intuitive and interpretable but often fail to capture semantic ambiguities or deeper structural complexities. The second category consists of global metrics, such as mean perplexity or semantic similarity measures. While these methods provide comprehensive quantitative evaluations, they frequently lack interpretability and connection to underlying local textual properties. This divide between local interpretability and global comprehensiveness reflects the broader challenge of bridging microscopic (token-level) and macroscopic (long-range, structural) perspectives of LLM behavior.

In this work, we propose *correlation dimension*, a measure drawn from fractal geometry and dynamical systems theory, to bridge this gap. Correlation dimension quantifies self-similarity—a

39th Conference on Neural Information Processing Systems (NeurIPS 2025).

fundamental characteristic of complex systems that exhibit invariant patterns across scales. Originally developed for analyzing deterministic chaotic systems [20], correlation dimension has since been successfully adapted to stochastic processes and real-world complex phenomena [45, 31]. Applied to language models, correlation dimension effectively quantifies the intrinsic complexity and recurrence structure of generated texts as perceived by the model. For instance, texts with randomly shuffled words appear highly complex and yield high dimensionality, whereas simple repetitive patterns exhibit low dimensionality.

We specifically propose computing correlation dimension using sequences of next-token log-probability vectors, which are readily available for any autoregressive language model. Recurrences are defined via the Euclidean distance between these vectors. Unlike perplexity, correlation dimension reflects deeper structural properties of the text generation process. For example, a model experiencing mode collapse might produce contextually irrelevant but statistically plausible text, exhibiting low perplexity yet high correlation dimension. Conversely, models lacking adequate memory to handle long-range dependencies might show low dimensionality despite high perplexity, clearly indicating their limited structural comprehension.

Correlation dimension serves as both a practical and theoretically grounded metric. It requires little computational overhead beyond perplexity calculations and runs at inference time, making it easy to integrate into existing inference infrastructures such as `vllm`. At a theoretical level, it provides nuanced insights into model behavior, revealing the long-range complexity structures that perplexity alone cannot uncover. Practically, it offers a straightforward yet powerful method to evaluate model robustness, indicate potential problems such as hallucination, and capture subtle forms of degeneration such as incoherence, and blandness.

Throughout the paper, we illustrate how correlation dimension inherently connects local recurrence structures and global textual complexity (Section 3), provides insights into intrinsic properties of texts and model behavior (Section 4), and effectively detects degeneration issues in long-text generation (Section 5). Overall, our approach naturally integrates interpretability and comprehensiveness, offering a robust metric grounded in the fundamental properties of complex dynamical systems.

## 2   Related Works

### 2.1   Statistical Self-Similar Phenomenon in Language

Self-similarity is a fundamental property observed in various complex systems, characterized by invariant patterns across multiple scales. A special class of self-similarity, known as scale-free properties, refers specifically to patterns that are consistent when viewed at different scales [52]. Unlike precise scale-free structures defined mathematically (e.g., the Koch snowflake), *statistical self-similarity* denotes approximate scale invariance identified through statistical analysis of empirical data [49].

Statistical self-similarities have been widely documented in natural language, manifesting through phenomena such as Zipf's law [58, 37, 40] for word frequencies, Herdan-Heaps's law [26, 24] for vocabulary growth, and long-range correlations [15, 14, 4, 50, 9] for word occurrence or word rank across text spans.

Previous approaches have explored statistical self-similarity and fractal dimension at the abstract semantic level, extending beyond traditional lexical-based metrics (e.g., word frequency). For instance, [11, 12] utilized latent semantic analysis (LSA) embeddings of text paragraphs [10] and discovered geometric self-similarity (fractal structures) within semantic spaces, estimating fractal dimensions approximately between 8 and 20. [44] also observed fractal patterns in word vectors with a Box-counting dimension of about 20. These findings are based on static word vectors that do not vary with context, and therefore have limited utility for characterizing language models.

Recently, [13] investigated the self-similarity of texts using autoregressive LLMs, and found consistent semantic self-similarity across multiple languages. However, the linguistic significance of the self-similarity and the explicit role of language models in capturing such phenomena remains unexplored. In contrast, this paper proposes a new method tailored for analyzing LLMs, and provides a range of insights into the generative dynamics of LLMs.

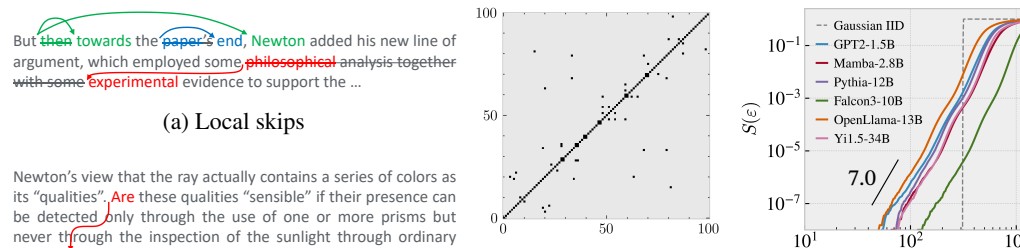

(a) Local skips

(b) Long-range skips (whole sentences)      (c) Recurrence plot      (d) Correlation integral curves

**Figure 1:** Experimental results on the "Newton's Philosophy" article of the Stanford Encyclopedia of Philosophy. (a-b) Examples of textual skips as predicted by the Pythia-12B model, illustrating both local and long-range recurrences in the text. (c) Segment of the recurrence plot for the log-probability vectors generated by the model; black dots indicate pairs of points within a specified distance threshold. (d) Correlation integral curves for six pre-trained language models (solid lines), compared to i.i.d. Gaussian noise in $\mathbb{R}^{50000}$ (dashed lines).

**Characterizing LLMs via Self-Similarity**     Previous studies have evaluated language models via statistical self-similarity. [47] compared generated texts from language models to natural texts across five scale-free properties, revealing significant deviations in scaling exponents. Building on this, [39] introduced quantitative statistical tests to measure such deviations systematically and examined how different sampling strategies (e.g., nucleus sampling vs. beam search) affect self-similarity. [6] set up a mutual-information scaling law for long-context language modeling.

Notably, [2] recently measured the fractal dimension and Hurst exponent of cumulative log-perplexity series, demonstrating that these scaling exponents correlate with downstream performance of LLMs. However, their approach primarily captures long-range dependencies in the model's predictive error sequences. In contrast, our method (detailed in Section 3) examines the intrinsic generative recurrence structure inherent in language itself, uncovering fundamentally different insights about the model behavior and text complexity.

## 3    Recurrence Structure and Correlation Dimension of Language

Natural language exhibits self-similarity across various linguistic scales, ranging from morphological, lexical, and syntactic to semantic structures and long-range dependencies. Unlike mathematically precise fractals, real-world self-similarity is typically approximate and statistical, manifesting through recurrent statistical patterns rather than exact repetitions.

The *correlation dimension* provides a quantitative framework to measure such statistical self-similarity in sequences by analyzing their *recurrence* structure. A recurrence is defined as an event where the trajectory of a system approximately revisits a previous state within a predefined distance threshold $\varepsilon$. As $\varepsilon$ increases, more recurrences naturally emerge. Self-similar systems typically follow a power-law relationship between the recurrence frequency $S(\varepsilon)$ and the distance threshold $\varepsilon$, formally defined as follows:

**Definition 1** (Correlation dimension [20])**.** *Given an infinite sequence $\{x_t\}_{t=1}^{\infty}$ in a metric space (e.g., $\mathbb{R}^D$), its correlation dimension $d$ is the exponent characterizing the scaling behavior of the correlation integral $S(\varepsilon)$:*

$$S(\varepsilon) \quad \propto \quad \varepsilon^d \quad as \quad \varepsilon \to 0, \tag{1}$$

*where the correlation integral $S(\varepsilon)$ is defined as the frequency of point pairs separated by distances less than $\varepsilon$:*

$$S(\varepsilon) = \lim_{t \to \infty} \frac{2}{t(t-1)} \sum_{1 \le i < j \le t} 1\{\|x_i - x_j\| < \varepsilon\}. \tag{2}$$

*Here, $\|\cdot\|$ denotes the Euclidean norm, and $1\{\cdot\}$ is the indicator function that equals 1 if the condition is true and 0 otherwise.*

Correlation dimension was originally developed to characterize deterministic attractors (e.g., Hénon maps [25]) but subsequently generalized for analyzing stochastic processes such as fractional Brownian motion [17] and complex networks [31].

**Correlation Dimension Applied to Language.** Representing text as a sequence of numerical vectors allow the identification and measurement of recurrences and subsequently their correlation dimension. However, selecting a meaningful, stable, and interpretable representation for natural language is nontrivial.

We propose utilizing the Euclidean distance between *logarithmic* next-token probability vectors derived from language models. Specifically, the log-probability vector at time $t$, denoted $x_t$, is calculated as:

$$x_t(\omega) = \log P_\theta(\omega_t = \omega | \omega_{t-c}, \cdots, \omega_{t-1}) \quad \forall \omega \in \mathbf{\Omega}, \tag{3}$$

where $\omega_t$ represents the token at position $t$, $P_\theta$ represents the model-predicted probability ($\theta$ specifies the model), and $\mathbf{\Omega}$ is the vocabulary set. Unless otherwise specified, the context length $c$ is set to infinity, i.e., no context length limit; its impact is further examined in Section 4.2 and discussed in Appendix A.

**Textual Skips as Recurrences.** Recurrences in next-token log-probability vectors can be interpreted as *textual skips*. If two states $x_t$ and $x_s$ ($s < t$) are close, the text segment $[s, t)$ could theoretically be omitted without significantly altering subsequent text generation. Skips occur at multiple scales, as illustrated in Figure 1(a-b): from local (single-word skips) to global (sentence-level skips). Smaller distance thresholds ($\varepsilon$) identify local skips, while larger thresholds detect longer-range skips. The hierarchy aligns naturally with Chomsky's generative grammar [7], where such skips may correspond to omitted subtrees within the hierarchical structure of language.

Figure 1(c) provides an example recurrence plot based on log-probability vectors from the text *Newton's Philosophy* [29], clearly visualizing recurrences at a certain threshold. The correlation integrals $S(\varepsilon)$, illustrated in Figure 1(d), demonstrate near-linear scaling for multiple pre-trained language models (GPT2 [43], Pythia [5], Falcon3 [51], OpenLLaMA [19], Yi1.5 [54], and Mamba [23]), yielding a consistent correlation dimension around 7. In contrast, Gaussian random vectors in equivalent dimensional spaces exhibit distinct different scaling behavior, underscoring language-specific recurrence structures.

## 3.1 Sufficiency of Next-Token Log-probabilities

The next-token log-probability vectors represent only partial information of a language model's full state, defined theoretically as the distribution over all future token sequences. Thus, one might question the sufficiency of next-token probabilities alone for characterizing long-range text generation.

Time-delayed embeddings, motivated by Takens' embedding theorem [48] and its stochastic extensions [45, 46], are common methods for reconstructing full states from partial observations. Such embeddings concatenate multiple sequential log-probability vectors, $y_t = [x_t; x_{t+1}; \ldots; x_{t+k}]$, potentially capturing higher-order dependencies.

However, empirical observations reveal negligible differences in the correlation dimension when comparing simple next-token log-probability vectors ($k = 1$) to embeddings with higher orders ($k > 1$), after accounting for the noise inevitably introduced in the delayed embeddings. This somewhat surprising result suggests that single-step log-probability vectors inherently encode significant long-term structural information about language evolution. This phenomenon mirrors findings in knowledge distillation [27], where single-step probability distributions effectively summarize model knowledge. Additional details and analysis are provided in Appendix E.

## 3.2 Empirical Convergence of Correlation Dimension

Considering an ideal "perfect" language model capable of precisely predicting next-token distributions, its correlation dimension would reflect intrinsic textual complexity. However, slight imperfections in predictions significantly affect correlation dimension measurements, particularly due to rare outcomes influencing the measure disproportionately compared to typical loss calculations. Precisely, let $l(\theta) = \mathbb{E}_{\omega_{\leq t} \sim \mathbb{P}} \left[ \log P_\theta(\omega_t | \omega_{\leq t-1}) \right]$ be the cross-entropy loss function of a language model parameterized by $\theta$, where $\mathbb{P}$ and $P_\theta$ represent the empirical distribution and the model-predicted distribution, respectively, and $\omega_{\leq t} = [\omega_1, \cdots, \omega_t]$ represents the sequence of tokens until time $t$. Then, we can see that the contribution of rare outcomes $\omega$ to the total loss is very small, proportional

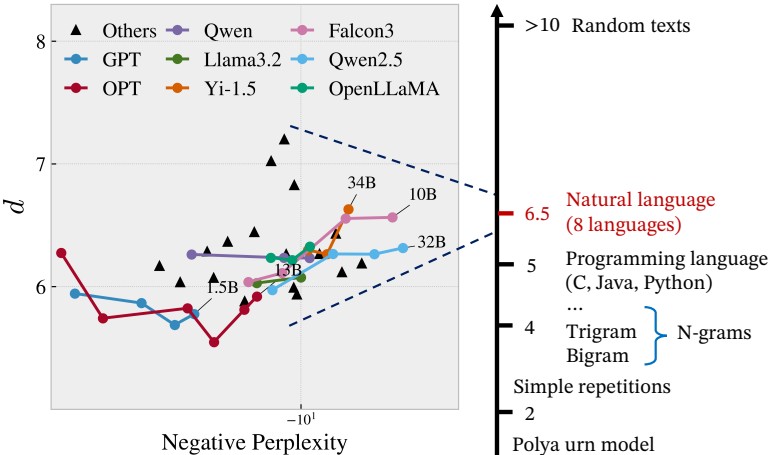

**Figure 2:** Left: Mean correlation dimension of the SEP dataset measured with various pre-trained LLMs. Right: A conjectured spectrum of correlation dimension for different types of texts / language models.

to its frequency:

$$\frac{\partial\, l(\theta)}{\partial\, \log \mathrm{P}_\theta(\omega|\omega_{\leq t-1})} = \mathbb{P}(\omega|\omega_{\leq t-1}) \qquad \forall \omega \in \boldsymbol{\Omega}. \tag{4}$$

In other words, the log-probability vectors at rare words $\omega$ may vary significantly, without affecting the loss function, but this variation will be reflected in the correlation dimension regardless of the rareness of the word.

Nonetheless, empirical experiments indicate convergence in correlation dimensions of well-trained language models to a narrow range as perplexity decreases. Figure 2 illustrates correlation dimension for various pre-trained LLMs across the Stanford Encyclopedia of Philosophy (SEP) [55] dataset which is summarized in Appendix C. As the perplexity decreases, correlation dimensions stabilize around a consistent value near 6.5. This empirical convergence underscores correlation dimension's reliability as a stable, interpretable metric for evaluating the structural complexity of natural language and the performance of language models.

The right half of Figure 2 presents the spectrum of correlation dimension values across various types of texts and statistical processes. Randomly shuffled texts exhibit high correlation dimensions, typically above ten, whereas self-reinforcing processes such as the Polya urn model [35] display much lower values, below two. $n$-gram processes with small $n$ also yield lower correlation dimensions compared to natural language. Moreover, programming languages (Python, Java, and C) show a consistent correlation dimension around 5. We provide the full results in Appendix C.2 (other natural languages) and C.3 (programming languages).

## 4 Characterizing Language Models Using Correlation Dimension

Estimating the correlation dimension of a text using language models raises critical questions regarding the reliability and interpretability of this measure when the underlying model is imperfect or insufficiently trained. Ideally, with a perfect language model, the correlation dimension would represent the intrinsic complexity of the text itself. However, real-world language models inevitably contain imperfections, prompting us to investigate whether the correlation dimension remains meaningful under such conditions.

In this section, we demonstrate that the correlation dimension is a robust measure of the *perceived complexity* of texts by LLMs, even when the LLMs are imperfect or insufficiently trained. Specifically, we explore three key aspects: (1) how the correlation dimension reflects the intrinsic complexity of a text as determined by its underlying hierarchical structure (Section 4.1); (2) how the correlation dimension is influenced by constraints on the contextual information available to the model (Section 4.2), and (3) how three distinct stages emerge in the training process of LLMs as indicated by the correlation dimension, a phenomenon not observed in perplexity (Section 4.3). After that, we present

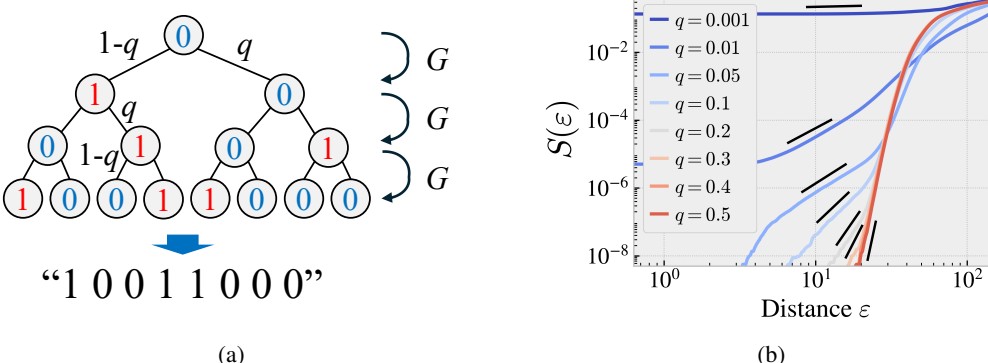

(a)                                                    (b)

**Figure 3:** (a) Illustration of the Lin-Tegmark grammar generation process governed by the transition matrix $G$, parameterized by $q$. (b) Correlation integral curves for texts generated with different $q$ values, measured using the OpenLLaMA-13B model.

a case study of occasions where significant divergence of correlation dimension is observed between different models on knowledge-intensive texts, and how this divergence indicates LLMs' hallucination behavior (Section 4.4).

## 4.1 Correlation Dimension and Textual Complexity

To analyze how correlation dimension captures inherent textual complexity, we generate sequences using the Lin-Tegmark grammar [33], a probabilistic context-free grammar defined over a binary alphabet $\{0, 1\}$. The grammar's complexity is parameterized by a single Bernoulli parameter $q \in [0, 1]$, as shown in Figure 3(a), controlling the mutual information decay between tokens: $G = \begin{bmatrix} q & 1-q \\ 1-q & q \end{bmatrix}$, where $G_{ij}$ denotes the probability of generating a child node valued $j$ given a parent node valued $i$. The resultant text complexity varies from highly predictable sequences (as $q$ approaches 0 or 1) to highly unpredictable sequences (as $q$ approaches 0.5).

Figure 3(b) illustrates the correlation integral curves computed by the OpenLLaMA-13B model for different values of $q$. As $q$ increases towards 0.5, the correlation dimension increases markedly from near zero to values above ten, demonstrating its sensitivity to textual complexity. Despite the nonlinear curves due to the model not being explicitly trained on this grammar, the correlation dimension consistently captures the inherent complexity defined by the grammar.

## 4.2 Effect of Contextual Constraints on Correlation Dimension

We next explore how limiting contextual access affects the correlation dimension measured by pretrained language models. Varying the context length parameter $c$ in Eq. (3) imposes restrictions on the model's available context, directly influencing its complexity perception. A context length of $c = 1$ reduces the model effectively to a bigram approximation, while longer contexts progressively enable deeper linguistic comprehension.

Figure 4(a) shows correlation integral curves for the Pythia-1B model measured on the SEP dataset at varying context lengths. Figure 4(b) depicts average correlation dimensions against perplexity for multiple models (Pythia-1B [5], Qwen2.5-1.5B [53], and Llama3.2-1B [21]). Notably, we observe a two-stage pattern: initial correlation increases sharply from approximately 3 to about 8 as context length extends to 32 tokens, followed by a gradual reduction to around 6.5 at longer contexts.

This pattern indicates that initial increases in context length enhance the model's perception of complexity, revealing more contextual variation. Subsequently, the model identifies redundant contextual variations, compressing perceived complexity and converging to a stable dimension around 6.5. This trend consistently appears across different models, validating the correlation dimension as a reliable metric of the complexity perceived by language models.

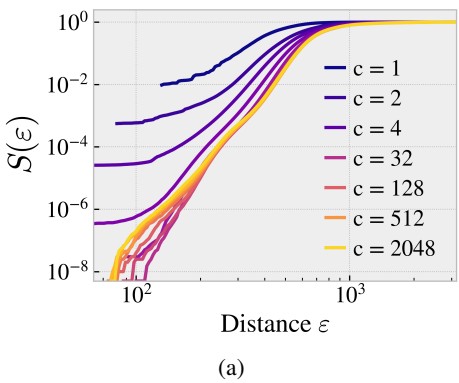
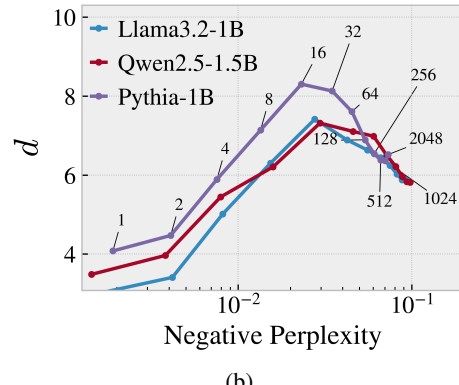

(a)

(b)

**Figure 4:** (a) Correlation integral curves on an article from the SEP dataset, measured by Pythia-1B model at different context lengths. (b) Mean correlation dimension on the SEP dataset (vertical axis), measured using three models at different context lengths, with respect to the negative perplexity (horizontal axis).

## 4.3 Three-Stage Evolution in LLM Pre-Training

Pre-training language models typically focus on monotonically reducing perplexity, implying a linear improvement. However, we find that correlation dimension reveals a distinct, nonlinear evolution consisting of three stages during training.

We monitored correlation dimension across multiple checkpoints of various models, including the Pythia family (12B, 2.8B, 1B, 160M, 14M), OpenLLaMA (3B, 13B), and Amber-7B (Figure 5). The evolution distinctly unfolds in three stages: (1) an initial rapid drop in correlation dimension due to learning short-range (bigram-level) structures, (2) a subsequent increase as models begin capturing longer-range dependencies, and (3) a final gradual decline indicating improved generalization via context compression. This three-stage evolution is clearly observed in the Pythia family, for which the early-stage checkpoints are available; for the other models, the evolution is verified in the last stage. Notably, these shifts occur even though perplexity continuously decreases.

Interestingly, the third stage, indicative of improved generalization, does not universally occur. Smaller models, like Pythia-14M and -160M, instead exhibit a sudden increase in correlation dimension at later stages, correlating with degraded performance in contextual learning tasks. Near the end of the training process, the correlation dimensions of these models exhibit a sudden rise to around 8. In the lower half, we show the accuracy of the two models in a simple in-context learning task, where the model is asked to repeat a sequence of symbols. A clear correlation is seen between the accuracy and the increase in correlation dimension, suggesting the loss of generalization ability of the model. This observation underscores the potential of correlation dimension to indicate generalization failures and stability in language models, insights inaccessible through perplexity alone.

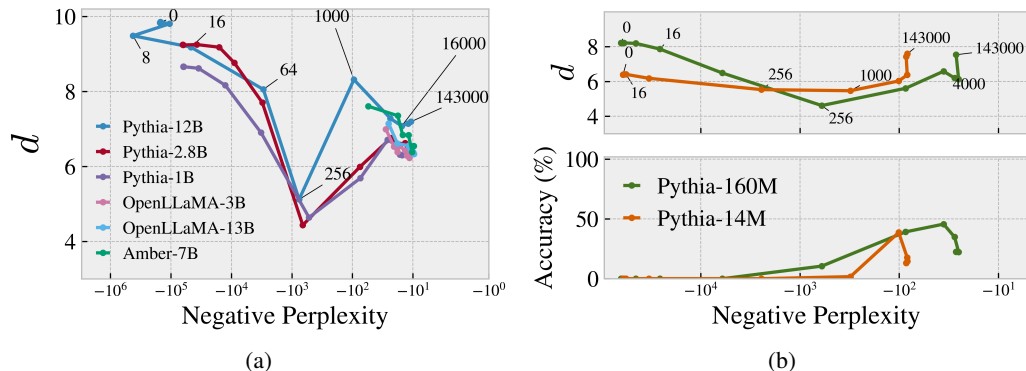

(a)

(b)

**Figure 5:** (a) Evolution of the mean correlation dimension for various language models on the SEP dataset during pre-training. (b) Notable case of two smaller models (Pythia-14M and 160M) that show a marked increase in correlation dimension (top) near the end of training, coinciding with a significant drop in in-context learning accuracy (bottom).

### 4.4 Hallucination vs. Memorization: Diverging Correlation Dimension on Knowledge-Intensive Texts

When processing texts containing domain-specific knowledge, LLMs may exhibit markedly different correlation dimensions, depending on whether the model truly recalls the knowledge or hallucinates [30]—producing syntactically valid but factually incorrect text. A model that successfully retrieves knowledge from memory tends to show a higher correlation dimension, whereas a model that hallucinates typically exhibits a lower correlation dimension on such texts.

We present a case study using the SEP article "process-theism," which contains a long list of relatively obscure scholars' names. Table 1 compares the correlation dimensions of the Qwen2.5 and Falcon3 model families on the entire SEP dataset (second column) and on the specific "process-theism" text (third column). Furthermore, we asked each model to complete the name list in the text and assessed whether the model successfully recalled the correct names or hallucinated; the results are summarized in the fourth column (see Appendix C.4 for details).

As shown in the table, Falcon3-7B (6.68) and Falcon3-10B (8.49) exhibit substantially higher correlation dimensions than the other models. Model size alone is not the determining factor: although Qwen2.5-32B has far more parameters, its correlation dimension remains low (4.42). A clear relationship emerges between correlation dimension and the model's ability to recall the correct names—every model with a correlation dimension below 5.0 produced hallucinations in this task.

**Table 1:** Comparison of correlation dimension between Qwen2.5 and Falcon3 on the text "process-theism".

| Model | Normal text (ave.) | Knowledge-intensive text | Recalling or Hallucinating |
|---|---|---|---|
| Qwen2.5-0.5B | 5.88 | 3.32 | hallucinate |
| Qwen2.5-7B | 6.27 | 3.56 | hallucinate |
| Qwen2.5-32B | 6.32 | 4.42 | hallucinate |
| Falcon3-1B | 6.03 | 3.28 | hallucinate |
| Falcon3-3B | 6.11 | 3.14 | hallucinate |
| Falcon3-7B | 6.55 | 6.68 | recall |
| Falcon3-10B | 6.56 | 8.49 | recall |

These results suggest that LLMs may internally signal whether they are hallucinating, and that correlation dimension provides a quantitative measure of this tendency. A possible interpretation is that recalling factual names requires the model to engage long-range dependencies, resulting in a high correlation dimension. In contrast, hallucination relies primarily on format-level imitation, leading to a markedly lower correlation dimension.

## 5 Correlation Dimension for Quantifying Degeneration in Text Generation

A critical challenge in language model generation is maintaining coherence, diversity, and relevance throughout extended text sequences. A well-known phenomenon, termed *degeneration*, describes scenarios where texts become repetitive, incoherent, or bland [28]. While explicit repetition is easily detectable, subtler forms of degeneration such as incoherence or blandness lack clear definitions or reliable quantification methods.

In this section, we propose using correlation dimension as a unified metric to quantify degeneration. Conceptually, degeneration is viewed as a sudden collapse from a higher-dimensional trajectory in the model's state space into a lower-dimensional attractor. Such collapses are generally irreversible, mirroring the *boundary crisis* phenomenon in chaotic dynamical systems [22]. Table 2 compares several popular evaluation methods (rows) against specific degeneration types (columns),

**Table 2:** Schematic summary of various LLM evaluation methods (rows) and their ability to detect different types of degeneration (columns). Triangles indicate potential usefulness that has not been empirically validated.

| Metric | Repetition | Incoherent | Bland |
|---|---|---|---|
| *Local generation probability* | | | |
| Perplexity | △ | × | △ |
| Cond. Entropy | △ | × | △ |
| *Word or $N$-gram statistics* | | | |
| Zipf Coefficient [58] | ✓ | △ | × |
| Heap Coefficient [24] | ✓ | △ | × |
| Rep-N [28] | ✓ | × | × |
| Distinct-N [32] | ✓ | × | × |
| Self-BLEU [57] | ✓ | × | △ |
| *Semantics* | | | |
| BERTScore [56] | × | ✓ | △ |
| MAUVE [41] | × | ✓ | △ |
| CorrDim (ours) | ✓ | ✓ | ✓ |

highlighting correlation dimension's unique capability to detect all considered forms: repetition, incoherence, and blandness.

## 5.1 Repetition Detection: Semantic vs. Surface-level

Texts that exhibit repetitive patterns tend to have low correlation dimensions. As shown in Table 3 (upper half), the correlation dimensions of explicitly repetitive texts fall below 2.0—far lower than those of normal texts (around 6.5). Traditional repetition-detection metrics such as Rep-N can also identify these patterns through lexical statistics, as indicated in the rightmost column.

Unlike Rep-N, however, the correlation dimension measures a text's *semantic* complexity rather than its surface-level word repetition. To show this distinction, we conducted a case study using Japanese, where texts can be written in two parallel script systems: (1) kanji (Chinese characters) combined with kana (Japanese phonetic symbols), and (2) kana only. The latter can be derived from the former by replacing each kanji with its phonetic kana equivalent. Table 3 (lower half) shows that the correlation dimensions are highly consistent between the two scripts, even though the kana-only script has a vocabulary roughly ten times smaller. In contrast, the lexicon-based metric, Rep-N, shows large differences between the two.

**Table 3:** Correlation dimension for repetition detection, compared with a lexicon-based metric (Rep-N).

| Text | CorrDim (mean) | Rep-2 (mean) |
|---|---|---|
| Normal texts (SEP dataset) | 6.27 | 0.45 |
| Explicitly repetitive patterns | 1.83 | 0.99 |
| **10 Japanese novels** | | |
| Normal script (kanji + kana) | 6.44 | 0.60 |
| Syllabic script (kana only) | 6.57 | 0.78 |
| Mean relative difference | 5.7% | 29.8% |

These findings demonstrate that correlation dimension is insensitive to superficial morphological variations and instead captures the intrinsic semantic repetition of language.

## 5.2 Detecting Degeneration Beyond Repetition

In addition to repetition, correlation dimension can be used to detect other forms of degeneration which are more subtle. To demonstrate this, we created a controlled dataset comprising responses to twenty generic questions. Each question had ten normal responses and intentionally degenerate texts—repetitive, incoherent, or bland—generated by the GPT-4o model (details in Appendix D). We computed correlation dimensions using the Falcon3-10B model [51].

The results, shown in Table 4, indicate significantly lower correlation dimensions for degenerate texts compared to normal responses, confirmed by Wilcoxon signed-rank tests (all p-values < 0.01). These findings validate the correlation dimension's sensitivity to different degeneration modes, including those traditionally difficult to quantify, such as incoherence and blandness.

**Table 4:** Correlation dimension or perplexity of degenerate texts compared to normal texts, measured using the Falcon3-10B model, with p-values from Wilcoxon signed-rank test in the second column.

| Group | CorrDim | | Perplexity |
|---|---|---|---|
| | mean | p-value | mean |
| Normal | 5.04 | - | 10.79 |
| Repetitive | 3.80 | 9.5E-7 | 1.25 |
| Incoherent | 3.96 | 2.9E-6 | 13.24 |
| Bland | 4.51 | 1.1E-3 | 4.24 |

For comparison, the table also reports the mean perplexity (right-most column) of each group, as measured by the Falcon3-10B model. While perplexity can distinguish certain types of degeneration—show much lower values for repetitive texts and higher values for incoherent texts compared to normal response—the direction of change is inconsistent across degeneration types. This indicates that perplexity is not an intrinsic measure of degeneration.

## 5.3 Stress-Testing Language Models with Random Texts

To assess model susceptibility to degeneration, we designed a stress-test involving random texts—specifically, a list of randomly generated English names separated by commas, totaling approximately 1024 tokens. Models were tasked with completing the sequence from varying input lengths ($n$), generating $1024 - n$ tokens. Random sequences typically have high initial correlation dimension due to unpredictability; thus models prone to degeneration exhibit abrupt drops in dimension as repetition or incoherence emerges.

Figure 6 illustrates correlation dimension trends for three models as the input length $n$ increases (and the output length $1024 - n$ decreases). Robust models like Yi1.5-34B consistently increased their correlation dimension, whereas weaker models such as Qwen2-7B-Instruct displayed marked dimension drops, indicating degeneration.

Further quantification is presented in Table 5, which reports mean correlation dimensions at input length $n = 512$ alongside scores from the HelloEval text-completion benchmark [42]. A high correlation (Spearman's $\rho = 0.952$) is evident, underscoring correlation dimension's validity as an intrinsic measure of model robustness in generating long, coherent text without relying on complex evaluation tasks.

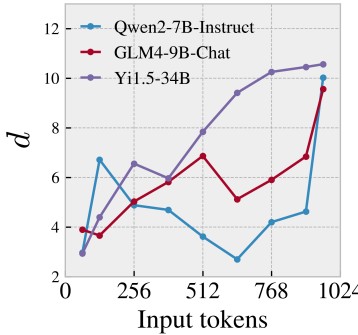

**Figure 6:** Correlation dimension of generated texts with respect to the length of input texts.

**Table 5:** Mean correlation dimensions of generated texts for each model, compared with the HelloEval text completion scores.

| Model | HelloEval | CorrDim |
|---|---|---|
| Qwen2-7B-Instruct | 5.12 | 3.54 |
| Llama3.1-8B | -5.61 | 4.15 |
| InternLM2.5-7B | 6.39 | 4.61 |
| Mistral-7B-v0.2 | 13.05 | 5.23 |
| GLM4-9B-Chat | 12.32 | 6.45 |
| LongWriter-GLM4-9B | 17.67 | 8.01 |
| InternLM2.5-20B | 36.68 | 8.89 |
| Yi1.5-34B | 44.73 | 8.89 |
| Spearman's $\rho$ | - | 0.952 |

## 6 Practical Issues

**Efficient Calculation** Computation of the correlation integral in Eq. (1) requires evaluating pairwise distances between log-probability vectors. This operation entails $O(N^2)$ additional memory and $O(N^2\Omega)$ computational cost, where $N$ ($\sim 10^4$) and $\Omega$ ($\sim 10^5$) denote the sequence length and vocabulary size, respectively. To reduce the cost, we employ two techniques—GPU kernel fusion and vocabulary reduction (see Appendix B)—which together achieve more than a $10\times$ speedup and incur **zero** additional memory overhead beyond standard LLM inference.

**Inference Precision** Modern LLMs typically operate at very low precision during inference to improve memory efficiency and throughput, with parameters often quantized to fewer than 4 bits. Remarkably, the correlation dimension remains highly stable even under such extreme quantization, provided that Euclidean distances between log-probability vectors are computed in FP32. In experiments with GPTQ [18] and AWQ [34]-quantized models, the average change in correlation dimension across the SEP dataset was below 3%. See Appendix B.3 for detailed results.

**Closed Models** Our approach requires access to an LLM's full logits, which is typically unavailable in closed models such as GPT-4. Nevertheless, because the calculation of correlation dimension relies solely on log-probabilities that are already produced during inference and requires no additional memory, we believe this measure can be readily integrated into standard commercial APIs.

## 7 Conclusion

We presented correlation dimension as a principled, model-agnostic metric for characterizing the long-range structure of texts as perceived by LLMs. By operating on next-token log-probability vectors, it bridges local recurrence with global complexity and runs efficiently at inference time. Empirically, correlation dimension (i) reflects intrinsic textual complexity, (ii) exhibits a predictable three-stage evolution during pre-training and under context constraints, and (iii) reliably detects degeneration (repetition, incoherence, blandness) beyond perplexity. These findings indicate that correlation dimension complements standard metrics by revealing aspects of model behavior that local accuracy alone misses. Future work includes formal analysis of estimation properties, extensions to conditional or multi-modal settings, and deployment as an online signal for training diagnostics and generation control.

## Acknowledgments

This work was supported by JST CREST, Japan, Grant Number JPMJCR2114.

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

# A   Settings for Estimating Correlation Dimension

While correlation dimension is defined for an infinitely long sequence, a finite number of steps are available in practice. For finite sequences, there is no guarantee that the correlation sum $S(\varepsilon)$ follows a power-law relationship with respect to $\varepsilon$. Nevertheless, we find that the correlation dimension measured from a finite sequence still provides an informative description of the complexity of the LLM and the underlying language dynamics.

The correlation dimension $d$ is obtained as the slope of $\log S(\varepsilon)$ versus $\log \varepsilon$, and the observable range of $\log S(\varepsilon)$, $\left[\frac{2}{N(N-1)}, 1\right]$, depends on $N$. To make $d$ comparable across sequences of different lengths, we limit the range of $S(\varepsilon)$ to $\left[\frac{20}{N(N-1)}, \frac{\eta}{N}\right]$ ($\eta = 1.0$ by default), i.e., we clip the left tail of the log-log plot below ten counts and discard the upper half. When the range becomes too narrow for small $N$ (i.e., $N < 500$), $\eta$ is increased to maintain a reasonable range for slope estimation.

**Short and Long Sequences**   We adopt slightly different settings for estimating the correlation dimension of long and short sequences. For long sequences, such as those in the SEP dataset, which exceed the context limit of typical LLMs, we use a moving-window approach to estimate next-token prediction probabilities with an exact, predefined context length (e.g., 512 tokens).

For short sequences, this approach would discard a large portion of the data. Therefore, we do not restrict the context length for short sequences. Since such sequences are typically within the model's context limit, the model can access all tokens at once. In this case, we input the entire sequence and compute the correlation dimension using the output probabilities at all time steps. Thus, the log-probability vectors are estimated with progressively increasing context lengths. This setting applies to the experiments in Section 5.

In practice, the difference between the two settings is often negligible, and we do not differentiate the two settings in the main text for simplicity.

**Synthetic Texts**   When measuring the correlation dimension of synthetic texts—specifically, the explicitly repetitive patterns described in Section 5.1—we further restrict the range of the distance threshold $\varepsilon$ to values above $10^1$ to prevent the correlation integral from being dominated by numerical errors. Because the synthetic texts follow exact repetitive patterns, the log-probability vectors lie very close to one another at small distance thresholds—typically below $10^1$ in a high-dimensional space (with dimension greater than $10^5$). At such small thresholds, numerical errors in high-dimensional space can accumulate and dominate the correlation integral.

# B   Efficient Calculation of Correlation Dimension

We introduce two techniques to enable efficient computation of the correlation dimension: (a) kernel fusion (Section B.1) and (b) vocabulary reduction (Section B.2). Kernel fusion achieves **zero additional memory footprint** beyond standard LLM inference and up to a **3x speedup**, particularly for long sequences. Vocabulary reduction decreases memory usage and computational cost by **over 10x**, with only a minor loss in accuracy. These two techniques can be combined to achieve both benefits simultaneously.

In addition, we examine the effects of model quantization on correlation dimension estimation (Section B.3). We find that the correlation dimension remains remarkably stable across different quantization methods, even at extremely low precision (e.g., 4 bits).

## B.1   Fused Calculation of Pairwise Distance and Correlation Integration

The computation of correlation dimension consists of two steps: (1) calculating pairwise distances between all log-probability vectors, and (2) counting the number of pairs within each distance threshold $\varepsilon$ to obtain $S(\varepsilon)$. Calculating the pairwise Euclidean distances for a sequence of length $N$ requires computing and storing a distance matrix of size $O(N^2)$ at high precision (e.g., FP32), which is prohibitively expensive for long sequences.

A straightforward approach to reduce memory usage is to divide the distance matrix into blocks and process them sequentially. The number of pairs with distances smaller than a threshold can be counted within each block and then accumulated to obtain the total count for the full distance matrix. However, after computing a block distance matrix, one still needs to copy it from high-speed local memory (i.e., SRAM) back to low-speed global memory (i.e., HBM or CPU memory) before performing step (2). When $N$ is large, this copy operation becomes a major bottleneck.

Inspired by FlashAttention [8], we propose fusing the two steps into a single CUDA kernel. After computing each block distance matrix, the number of pairs within a distance threshold $\varepsilon$ is directly computed *in-place*, without copying (blocks of) the large distance matrix out of SRAM.

---

**Algorithm 1** Fused Blockwise Distance-and-Count for Correlation Integral

---

**Require:** Log-probability matrix $X \in \mathbb{R}^{N \times D}$, thresholds $\{\varepsilon_k\}_{k=1}^{K}$
**Ensure:** Counts $S(\varepsilon_k)$ for all $k$
1: Initialize global counters $S_k \leftarrow 0$
2: **for** each index tile $(i, j)$ with $j \leq i$ **do**
3:     Load $X_i$ and $X_j$ tiles into SRAM/shared memory
4:     Compute pairwise distances $d_{ij}$ on-the-fly within the tile
5:     Immediately compare $d_{ij}$ with $\{\varepsilon_k\}$ and perform `AtomicAdd` to $S_k$
6:     **Do not** write the tile distances $d_{ij}$ back to global memory
7: **end for**
8: **return** $\{S_k\}$

---

**Table 6:** Runtime comparison between our method (bottom row) and four baselines on a log-probability vector sequence of length 50,000 and dimension 10,000. Distance computation is performed in FP32 precision. For the blockwise baseline, we use a block size of 512×512.

| Methods | Additional memory | Clock time (s) | Speedup |
|---|---|---|---|
| `torch.pdist` (entire matrix at once) | 4.8 GiB | 44.3 | 0.07x |
| `torch.cdist` (entire matrix at once) | 9.3 GiB | 3.3 | 1.0x |
| `torch.cdist` (blockwise) | 1.0 MiB | 6.9 | 0.48x |
| `torch.cdist` (blockwise, upper triangular only) | 1.0 MiB | 3.6 | 0.92x |
| Fused (our method) | 0 | 1.8 | 1.83x |

Table 6 presents an empirical comparison between our method and four baselines using off-the-shelf PyTorch implementations. By fusing the two steps into a single CUDA kernel, our method achieves nearly a 2x speedup over the fastest baseline, while requiring no additional memory.

## B.2 Vocabulary Reduction

A key property of fractal dimension is its almost-sure invariance under linear projection.

**Theorem 2** (Fractal Projection Theorems [38, 16]). *Consider a fractal set embedded in $\mathbb{R}^D$ with Hausdorff dimension (or box-counting dimension) $d$ ($\leq D$). Given a random linear projection from $\mathbb{R}^D$ to $\mathbb{R}^m$ ($m < D$), then with probability 1, the projected set's Hausdorff dimension $\tilde{d}$ satisfies:*

$$\tilde{d} = \begin{cases} d, & \text{if } m \geq d, \\ m, & \text{if } m < d. \end{cases}$$

In other words, we can construct a linear projection that maps the log-probability vectors in $\mathbb{R}^\Omega$ to lower-dimensional vectors, and estimate the correlation dimension using the reduced vectors. However, the theorem assumes infinitely long sequences of points, which real data do not satisfy. Therefore, the choice of linear projection must be handled carefully.

We propose a simple modulo-based function to group dimensions (i.e., unique tokens) in $\mathbb{R}^\Omega$ and sum the vectors within each group to form a smaller vector. The linear projection $\psi_v : \mathbb{R}^\Omega \to \mathbb{R}^v$ is defined as:

$$\psi_v(\mathbf{x})_i = \sum_{j \in \text{mod}_v^{-1}(i)} x_j, \tag{5}$$

where $\mathbf{x} = [x_1, \cdots, x_\Omega]$ is a log-probability vector, $\psi_v(\mathbf{x})_i$ denotes the $i$-th element of the reduced vector, and $\mathrm{mod}_v^{-1}(i)$ is the set of indices $j$ such that $j \bmod v = i$. In practice, the vocabulary size $\Omega$ is approximately $10^5$.

Table 7 shows an example using the "newton-philosophy" article from the SEP dataset, where we measured the correlation dimension using the Qwen2.5-7B model with unlimited context length. As shown, using $v = 10{,}000$ achieves values very close to the true correlation dimension, while reducing memory and computation costs by roughly 10x.

Although randomly selected linear projections are sometimes recommended to avoid zero-probability exceptions in Theorem 2, we find that such randomness introduces uncertainty and significant bias in correlation dimension estimates when the vocabulary size $\Omega$ is large. Therefore, we prefer the deterministic modulo function defined in Eq. (5).

**Table 7:** Change in correlation dimension after vocabulary reduction.

| Dimension | CorrDim |
|---|---|
| 151643 (=$\Omega$) | 5.92 |
| 30000 | 5.79 |
| 10000 | 5.81 |
| 3000 | 5.87 |
| 1000 | 6.38 |
| 300 | 6.77 |

### B.3 Effects of Model Quantization

Although the correlation dimension is defined in the limit as $\varepsilon \to 0$, we observe that it remains remarkably stable under model quantization, including AWQ [34] and GPTQ [18], which compress model parameters to 4 bits. Table 8 reports the correlation dimensions estimated using quantized models for 60 articles in the SEP dataset. The second to fourth columns present the correlation dimensions obtained with the original (FP16), AWQ-, and GPTQ-quantized versions of the Qwen2.5-32B-Instruct model. As shown, both quantization methods yield only minor variations in correlation dimension, with a mean absolute change of just 0.14 across all articles.

**Table 8:** Effects of model quantization on correlation dimension estimation. For each article in the SEP dataset, we measured the correlation dimension of the first 10,000 tokens using the Qwen2.5-32B-Instruct model with unlimited context length.

| Text | FP16 | AWQ | GPTQ |
|---|---|---|---|
| aesthetics-18th-german | 6.06 | 6.21 | 6.06 |
| africana | 5.31 | 5.39 | 5.53 |
| ... | | | |
| trinity | 5.60 | 5.66 | 5.66 |
| weyl | 6.57 | 6.62 | 6.45 |
| Mean | 5.86 | 5.90 | 5.93 |
| - Mean Absolute Change | - | 0.14 | 0.14 |

# C   Supplementary Results for Sections 3 and 4

## C.1   English Texts

**The SEP Dataset**   We constructed the Stanford Encyclopedia of Philosophy (SEP) dataset by using the 60 longest articles on the SEP website [55]. The article identifiers are listed in the left-most column of Table 9. For each article, we removed the prologue and catalog. All 60 articles are longer than 20,000 words. When measuring the correlation dimension, we truncated each article to the first 20,000 tokens. This may introduce slight differences between models with different tokenizers, but the effect was small for most models we examined.

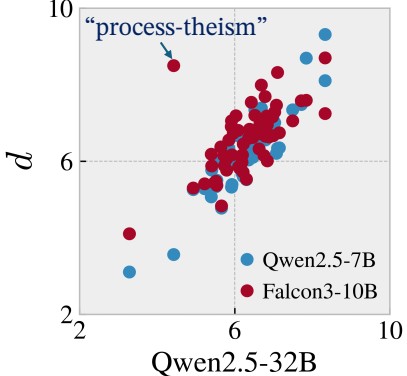

**Figure 7:** Comparison of correlation dimensions of the SEP articles across three models.

Results for three representative models—Qwen2.5-32B, Qwen2.5-7B, and Falcon3-10B—are shown in Table 9. The second to fourth columns report the articles' correlation dimensions; the right-most column lists perplexities for Qwen2.5-32B. Correlation dimensions exhibit strong consistency across models, especially within the same family (Qwen2.5), as illustrated in Figure 7.

The largest discrepancy in correlation dimension occurs for the article "process-theism", highlighted in Figure 7 and shaded in Table 9. This divergence reflects a qualitative behavioral difference between models on knowledge-intensive text: whether the model recalls knowledge or tends to hallucinate. We report this formally in Section 4.4, with additional details in Appendix C.4.

**Table 9:** Results on the SEP dataset acquired from three pre-trained LLMs. Context size is set to 512 tokens.

| Article | CorrDim | | | Perplexity |
| --- | --- | --- | --- | --- |
| | Qwen2.5-32B | Qwen2.5-7B | Falcon3-10B | Qwen2.5-32B |
| prisoner-dilemma | 3.28 | 3.11 | 4.10 | 3.58 |
| process-theism | 4.42 | 3.56 | 8.49 | 5.68 |
| cosmological-argument | 4.93 | 5.26 | 5.30 | 5.83 |
| information | 5.22 | 5.28 | 5.41 | 6.31 |
| ethics-chinese | 5.39 | 5.07 | 6.18 | 5.52 |
| recursive-functions | 5.40 | 5.76 | 5.88 | 4.96 |
| reasoning-automated | 5.48 | 5.32 | 5.44 | 5.06 |
| infinity | 5.51 | 5.33 | 5.48 | 5.68 |
| buddhism-tiantai | 5.53 | 5.50 | 5.37 | 9.25 |
| evil | 5.64 | 6.01 | 6.37 | 4.69 |
| logic-temporal | 5.66 | 4.77 | 4.83 | 4.99 |
| mohist-canons | 5.72 | 6.06 | 6.14 | 3.75 |
| linguistics | 5.73 | 6.00 | 5.88 | 9.54 |
| publichealth-ethics | 5.79 | 5.96 | 5.77 | 8.87 |
| mill-moral-political | 5.79 | 6.12 | 6.06 | 4.38 |
| gasset | 5.85 | 6.33 | 6.55 | 11.22 |
| computational-complexity | 5.91 | 5.95 | 6.90 | 5.32 |
| africana | 5.91 | 5.32 | 7.05 | 12.21 |
| principia-mathematica | 5.93 | 5.39 | 6.17 | 5.36 |
| levinas | 5.96 | 6.14 | 6.73 | 15.92 |
| formal-belief | 6.03 | 5.99 | 7.19 | 5.82 |
| spacetime-singularities | 6.09 | 6.66 | 6.81 | 5.42 |
| computational-linguistics | 6.10 | 6.01 | 5.85 | 9.71 |
| chaos | 6.13 | 5.84 | 5.81 | 6.70 |
| shaftesbury | 6.17 | 5.93 | 6.14 | 8.62 |
| possibilism-actualism | 6.18 | 5.68 | 5.98 | 6.72 |
| trinity | 6.19 | 6.63 | 6.16 | 7.97 |
| dynamic-epistemic | 6.21 | 5.70 | 5.71 | 5.44 |
| proof-theory | 6.22 | 5.60 | 6.45 | 6.36 |
| logics-for-games | 6.30 | 5.52 | 5.53 | 9.92 |
| game-theory | 6.36 | 6.03 | 6.60 | 3.84 |
| consciousness-temporal | 6.38 | 6.13 | 6.83 | 8.63 |
| sidgwick | 6.40 | 6.55 | 6.78 | 10.32 |
| epistemic-game | 6.42 | 6.31 | 7.54 | 5.43 |
| innateness-language | 6.47 | 6.10 | 6.62 | 4.98 |
| newton-philosophy | 6.51 | 6.63 | 6.75 | 6.19 |
| habermas | 6.52 | 7.34 | 7.20 | 9.04 |
| descartes-epistemology | 6.62 | 6.56 | 6.32 | 6.48 |
| russell-moral | 6.65 | 6.62 | 6.98 | 7.04 |
| mereology | 6.67 | 6.96 | 6.71 | 7.71 |
| connectives-logic | 6.68 | 7.39 | 7.99 | 6.63 |
| weyl | 6.75 | 6.87 | 6.15 | 5.62 |
| consciousness-intentionality | 6.78 | 6.62 | 7.69 | 9.57 |
| attention | 6.79 | 6.56 | 6.65 | 7.10 |
| margaret-cavendish | 6.81 | 6.81 | 6.75 | 8.16 |
| qm-action-distance | 6.83 | 7.12 | 7.18 | 4.70 |
| normativity-metaethics | 6.84 | 7.06 | 6.01 | 5.63 |
| epistemology-bayesian | 6.84 | 6.94 | 6.95 | 5.94 |
| aesthetics-18th-german | 6.88 | 6.89 | 6.63 | 7.75 |
| logic-inductive | 6.97 | 6.76 | 7.16 | 5.10 |
| chance-randomness | 7.00 | 7.07 | 6.67 | 6.88 |
| idealism | 7.03 | 7.01 | 7.29 | 6.94 |
| egalitarianism | 7.07 | 6.20 | 7.46 | 3.14 |
| reid | 7.10 | 6.34 | 8.32 | 5.43 |
| early-modern-india | 7.15 | 6.35 | 6.74 | 4.70 |
| heidegger-aesthetics | 7.49 | 7.34 | 7.05 | 7.42 |
| pythagoreanism | 7.71 | 7.48 | 7.58 | 4.10 |
| heidegger | 7.84 | 8.69 | 7.59 | 6.59 |
| al-farabi-metaphysics | 8.33 | 9.31 | 8.70 | 6.04 |
| ontological-commitment | 8.33 | 8.10 | 7.25 | 5.19 |
| Mean | 6.31 | 6.27 | 6.56 | 6.71 |
| Pearson's $\rho$ (w.r.t. CorrDim of Qwen2.5-32B) | - | 0.90 | 0.63 | -0.01 |

## C.2 Other Natural Languages

In addition to English, we conducted experiments on seven other natural languages: French, German, Spanish, Italian, Dutch, Chinese, and Japanese. Correlation dimension was measured using the multilingual Qwen2.5-7B model. For each language, we selected 10 books from Project Gutenberg. We used an unlimited context length and measured the correlation dimension on the first 10,000 tokens of each book.

Table 10 reports results for the eight languages. Each book is identified by its Project Gutenberg index, accessible at `https://www.gutenberg.org/ebooks/<book-id>`, where `<book-id>` is the index. A consistent dimension of about 6.0 is observed for normal texts in all languages, suggesting a language-independent, universal complexity across natural languages. Small variations between books are primarily attributable to genre and style. For example, the German selection includes multiple philosophical works by Hegel and therefore exhibits higher values than the Spanish selection, which is primarily novels.

The dimension values are slightly lower than those in Figure 2, which is a natural consequence of using an unlimited context length, as discussed in Section 4.2.

**Table 10:** Correlation dimensions of books in Project Gutenberg in different languages.

| English | | French | | German | | Spanish | |
|---|---|---|---|---|---|---|---|
| Book ID | CorrDim | Book ID | CorrDim | Book ID | CorrDim | Book ID | CorrDim |
| 74 | 5.75 | 9643 | 5.51 | 34811 | 5.74 | 2000 | 6.07 |
| 76 | 6.03 | 13819 | 6.18 | 44921 | 6.56 | 25640 | 5.82 |
| 9830 | 4.70 | 13952 | 6.33 | 14075 | 6.03 | 44584 | 5.18 |
| 86 | 6.63 | 13951 | 6.69 | 23756 | 6.29 | 33275 | 7.07 |
| 215 | 6.76 | 14158 | 7.16 | 8126 | 5.70 | 37590 | 5.51 |
| 37106 | 8.41 | 14163 | 5.40 | 6729 | 5.63 | 47092 | 3.41 |
| 805 | 5.08 | 796 | 6.17 | 6698 | 7.32 | 28281 | 5.28 |
| 996 | 5.52 | 14287 | 6.39 | 40739 | 4.95 | 49660 | 5.35 |
| 1156 | 5.90 | 9262 | 6.92 | 46259 | 5.74 | 17013 | 5.77 |
| 2701 | 6.08 | 5423 | 4.27 | 31114 | 7.31 | 14329 | 4.65 |
| Mean | 6.09 | | 6.10 | | 6.13 | | 5.41 |
| **Italian** | | **Dutch** | | **Chinese** | | **Japanese** | |
| Book ID | CorrDim | Book ID | CorrDim | Book ID | CorrDim | Book ID | CorrDim |
| 46957 | 5.61 | 19591 | 4.71 | 23835 | 6.26 | 31757 | 6.47 |
| 20062 | 5.90 | 25138 | 6.07 | 23950 | 6.18 | 34013 | 6.36 |
| 10215 | 5.00 | 13214 | 6.42 | 23910 | 5.72 | 34636 | 6.15 |
| 48490 | 5.87 | 19563 | 6.38 | 24226 | 5.55 | 35327 | 6.87 |
| 43022 | 6.18 | 19774 | 4.87 | 27582 | 6.52 | 37626 | 6.36 |
| 46082 | 4.27 | 21875 | 6.97 | 23962 | 5.80 | 32978 | 6.77 |
| 43023 | 6.19 | 17706 | 6.13 | 25350 | 5.50 | 33307 | 6.35 |
| 43024 | 6.19 | 27124 | 5.66 | 25142 | 6.01 | 36459 | 6.53 |
| 48445 | 5.21 | 19161 | 5.50 | 24264 | 5.49 | 32941 | 6.44 |
| 19024 | 5.89 | 26564 | 5.02 | 23841 | 6.26 | 31617 | 6.14 |
| Mean | 5.63 | | 5.77 | | 5.93 | | 6.44 |

## C.3   Programming Languages

We also conducted experiments on three programming languages: Python, Java, and C. For each language, we selected 30–50 sufficiently long source files from their standard libraries. We used the Qwen2.5-Coder-7B model to measure correlation dimensions. Because many source files are relatively short, we used an unlimited context length instead of a fixed one to avoid excessive truncation. In addition, we removed comments and docstrings to eliminate the influence of natural language content.

Results for the three programming languages are shown in Table 11. As shown, the correlation dimensions for all three languages are approximately 5.0—significantly lower than those of natural texts (around 6.5).

**Table 11:** Correlation dimensions of programs written in different programming languages: C (left), Java (middle), and Python (right).

| C | | Java | | Python | |
|---|---|---|---|---|---|
| Source code | CorrDim | Source code | CorrDim | Source code | CorrDim |
| nfsd_nfs4xdr.c | 6.27 | text_DecimalFormat.java | 5.58 | test_test_decimal.py | 3.85 |
| f2fs_super.c | 4.71 | util_Collections.java | 5.53 | test_test_io.py | 4.94 |
| btrfs_disk-io.c | 4.47 | math_MutableBigInteger.java | 4.23 | test_pickletester.py | 5.35 |
| ceph_mds_client.c | 4.08 | lang_Math.java | 5.34 | pickletools.py | 6.10 |
| f2fs_data.c | 4.56 | math_BigInteger.java | 5.71 | test__test_multiprocessing.py | 4.86 |
| f2fs_file.c | 4.96 | lang_FdLibm.java | 3.37 | test_datetimetester.py | 4.89 |
| ext4_namei.c | 5.70 | Character.java | 8.13 | tkinter___init__.py | 4.99 |
| btrfs_extent_io.c | 4.92 | util_Arrays.java | 4.22 | doctest.py | 5.15 |
| btrfs_send.c | 4.78 | text_SimpleDateFormat.java | 5.45 | test_test_inspect.py | 4.49 |
| btrfs_relocation.c | 4.55 | text_CompactNumberFormat.java | 4.92 | test_test_codecs.py | 4.08 |
| jfs_jfs_dmap.c | 5.08 | util_Formatter.java | 4.99 | email__header_value_parser.py | 6.56 |
| btrfs_inode.c | 5.10 | text_MessageFormat.java | 4.36 | test_test_dataclasses.py | 4.99 |
| ocfs2_refcounttree.c | 4.75 | lang_String.java | 6.13 | test_test_typing.py | 5.64 |
| resctrl_rdtgroup.c | 5.33 | util_HashMap.java | 4.54 | _pydecimal.py | 4.41 |
| namespace.c | 5.19 | util_ResourceBundle.java | 5.23 | test_test_enum.py | 4.61 |
| ocfs2_alloc.c | 4.69 | io_File.java | 5.12 | inspect.py | 6.08 |
| btrfs_ioctl.c | 4.89 | io_ObjectOutputStream.java | 3.94 | test_test_os.py | 4.64 |
| btrfs_free-space-cache.c | 5.15 | util_Calendar.java | 4.12 | test_test_subprocess.py | 4.67 |
| ocfs2_dlmglue.c | 4.52 | lang_System.java | 5.07 | test_test_doctest.py | 4.15 |
| nls_nls_cp950.c | 6.34 | security_KeyStore.java | 3.94 | pydoc_data_topics.py | 5.96 |
| dlm_lock.c | 4.37 | time_LocalDate.java | 5.71 | test_test_ssl.py | 3.94 |
| ext4_mballoc.c | 5.08 | lang_ClassLoader.java | 6.07 | test_test_descr.py | 4.45 |
| ntfs3_fslog.c | 4.92 | lang_AbstractStringBuilder.java | 5.47 | test_test_argparse.py | 6.16 |
| btrfs_extent-tree.c | 4.97 | util_Spliterators.java | 3.54 | test_test_zipfile.py | 4.22 |
| ceph_caps.c | 4.37 | io_ObjectInputStream.java | 5.71 | test_test_logging.py | 4.82 |
| btrfs_volumes.c | 5.01 | lang_Thread.java | 5.70 | turtle.py | 6.69 |
| btrfs_qgroup.c | 4.48 | net_URI.java | 4.32 | test_test_unicode.py | 5.55 |
| btrfs_block-group.c | 5.21 | util_Scanner.java | 4.38 | test_test_xml_etree.py | 4.67 |
| nfsd_nfs4state.c | 5.40 | util_GregorianCalendar.java | 4.70 | test_test_buffer.py | 3.95 |
| ext4_extents.c | 5.38 | time_ZonedDateTime.java | 4.76 | test_test_statistics.py | 5.28 |
| ocfs2_xattr.c | 4.60 | lang_StrictMath.java | 6.25 | pydoc.py | 5.60 |
| ext4_inode.c | 5.09 | util_DualPivotQuicksort.java | 5.54 | unittest_mock.py | 5.18 |
| nfs_nfs4xdr.c | 5.48 | util_JapaneseImperialCalendar.java | 4.65 | test_test_socket.py | 4.26 |
| btrfs_ctree.c | 5.44 | util_Locale.java | 4.79 | | |
| btrfs_tree-log.c | 4.99 | io_ObjectStreamClass.java | 4.95 | | |
| namei.c | 5.46 | math_BigDecimal.java | 3.75 | | |
| nfs_nfs4proc.c | 4.96 | lang_Character.java | 8.13 | | |
| jfs_jfs_dtree.c | 5.08 | util_TreeMap.java | 4.46 | | |
| nls_nls_cp949.c | 6.50 | lang_Class.java | 6.02 | | |
| ext4_super.c | 4.27 | | | | |
| ocfs2_dir.c | 4.71 | | | | |
| f2fs_segment.c | 4.73 | | | | |
| nls_nls_cp936.c | 6.49 | | | | |
| nls_nls_cp932.c | 6.12 | | | | |
| Mean | 5.07 | Mean | 5.10 | Mean | 5.00 |

## C.4   Hallucination vs. Memorization: Experimental Details

In Section 4.4 and Appendix C, we reported a notable divergence in correlation dimension between models on the knowledge-intensive SEP article "process-theism." The text contains an extensive list of philosophers' names, many of whom are relatively obscure. We conducted a simple experiment: we prompted the models to continue the name list and observed that those with lower correlation dimensions tended to hallucinate and generate non-existent names.

We began with the following input text and asked each model to complete the list:

> ... Philosophers and theologians who have published a monograph defending some variety of process theism informed by Whitehead or Hartshorne include: Henry Nelson Wieman , Bernard Meland , Paul Weiss , Norman Pittenger , Daniel Day Williams , John Moskop, William L. Reese, John B. Cobb, Jr., Schubert Ogden , Edgar A. Towne , Eugene H. Peters , Bowman Clarke , Joseph Bracken, Burton Z. Cooper, Marjorie Hewitt Suchocki, Gene A. Reeves, Lewis S. Ford , André Gounelle, Rem B. Edwards, **(start generation)**

The ground-truth continuation is:

> Delwin Brown , David A. Pailin, Franklin I. Gamwell, Forrest Wood, David Ray Griffin, James A. Keller, Jorge Luis Nobo , Tyron Inbody, Carol P. Christ, George L. Goodwin, Barry Whitney, Santiago Sia, Jay McDaniel, George W. Shields , Donald Viney, Catherine Keller, Daniel A. Dombrowski, Anna Case-Winters, Kurian Kachappilly, Gregory A. Boyd, Roland Faber, Thomas Jay Oord, Donna Bowman, Derek Malone-France, Monica A. Coleman, and Julia Enxing.

**Generation Results** We report generation results for seven models: Qwen2.5-0.5B, Qwen2.5-7B, Qwen2.5-32B, Falcon3-1B, Falcon3-3B, Falcon3-7B, and Falcon3-10B. Names that appear in the gold answer are marked in green. These names did *not* appear in the input context; hence, they are not inferred from context but recalled from the model's internal knowledge. All models were evaluated with a temperature of 1.0. We present representative generations that contain at least five names.

**Qwen2.5-0.5B** (CorrDim: 3.32):

> G. K. Dugald Stewart, Robert J. Leary, E. F. Robinson, Bernard P. Rieff, Robert A. Miller , R. K. Gombrich, Alan Tucker , J. B. Williams , J. V. Fowden , E. R. Brown , William E. Clark , David E. Stoloff , Eric Damer , James R. Miller , D. A. Burchardt , W. J.

**Qwen2.5-7B** (CorrDim: 3.56):

> Robert Schofield, Robert McAfee Brown, Keith Yandell, Robert John Russell, John H. Cobb, Jr., Arthur Roberts , Robert Solomon , William E. Connolly , and Bruce L. Reichenbach .

**Qwen2.5-32B** (CorrDim: 4.42):

> Eugene F. Rogers, Jr ., C. Robert Mesle, Keith R. Davies, William S. Hatcher , Roland Faber, Thomas J. J. Altizer, Mark McSwain , Charles Hartshorne, and David Ray Griffin.

**Falcon3-1B** (CorrDim: 3.28):

> Peter J. Haught, William B. McMillen, R. B. W. McMillen, Paul H. Metzger, Brian Joseph Murphy, George E. Spivey, Peter L. Wurm, Carl W. Smith, William H. Miller, and C. S. Lewis.

**Falcon3-3B** (CorrDim: 3.14):

> John C. Miller, Jr., Bernard Auchincloss, William L. McBride, Gerald L. T. Mitchell, George A. Meeks, William C. Schade, Thomas B. Allen, Jr., Michael W. Williams, David L. Boorse, John Henry Elliott, J. I. Packer, J. H. Oder, R. A. Butler, Bernard W. Miller, Robert E. Houtman, William L. Lane

**Falcon3-7B** (CorrDim: 6.68):

> David Ray Griffin, C. Robert Mesle, David R. Tracy, Robert C. Neville, Roland Faber, Jay McDaniel, and John D. Capes.

**Falcon3-10B** (CorrDim: 8.49):

> Delwin Brown, David Ray Griffin, Hank Keeton, George L. Robinson, George R. Reynolds, Franklin I. Gamwell, Alan I. Megill, Roland Faber, and Catherine Keller.

As shown, Falcon3-7B, and Falcon3-10B correctly recalled several names, whereas the other models generated plausible but non-existent ones. The Qwen2.5-32B model recalled two names correctly, but the two names are more well-known and easier to recall than the other names. Therefore, in avoiding

hallucination, model size alone is not decisive: Falcon3-10B achieved much higher precision than Qwen2.5-32B, despite being roughly three times smaller. We hypothesize that correlation dimension is a better indicator of a model's tendency to hallucinate in specific contexts—a low correlation dimension suggests that the model's generation collapses into a simple, format-driven pattern.

# D Repetition Detection: Experimental Details and Supplementary Results

## D.1 Repetition Detection

**Explicitly Repetitive Patterns have Low CorrDim $\approx 2$**   We generated texts by explicitly repeating a set of patterns and measured their correlation dimensions, as described in Section D.1. Results are shown in Table 12. The patterns to repeat are shown in the first column of the table, and the second column shows the mean correlation dimension of the repeated text. As shown, correlation dimensions of these explicitly repetitive patterns are about 1.5–2.5, significantly lower than those of normal texts (around 6.5).

A dimension close to 2 indicates that the LLM's internal state evolves like a random walk while retaining a steady memory of previous states.

A key consideration for correct estimation here is to restrict the measurement range to sufficiently large distance thresholds (Appendix A): for highly regular patterns, state distances become so small that numerical errors from high-dimensional log-probability vectors dominate. This adjustment is typically unnecessary for normal texts.

**Table 12:** Repetition detection on explicitly repetitive patterns.

| Repetition Pattern | Mean CorrDim |
|---|---|
| "01" | 2.17 |
| "012" | 2.29 |
| "ab" | 1.58 |
| "#%" | 1.94 |
| "#%@" | 1.73 |
| ")@#%^*!" | 1.79 |
| " 0 1" | 1.73 |
| " 0 1 2" | 1.87 |
| " a b" | 1.69 |
| " # %" | 1.71 |
| " # % @" | 1.71 |
| " ) @ # % ^ * !" | 1.71 |
| Mean | 1.83 |

**Japanese Scripts: CorrDim Invariance under Kanji–Kana Conversion**   We compared two Japanese script systems. The first is the standard, morphographic-plus-syllabic system, consisting of kanji (Chinese characters) and kana (Japanese phonetic symbols). In the second, all kanji are replaced by kana, so the entire text is written only in kana. The kana-only script has an order-of-magnitude smaller vocabulary.

We used the ten Japanese books listed in Table 10 from Project Gutenberg. Using the `kanjiconv` tool [1], we converted each book from the standard script to the syllabic (kana-only) script.

Table 13 reports the correlation dimensions (second and third columns) of the ten books in both scripts, computed with the Qwen2.5-7B model. The dimensions are highly consistent between the two scripts, even though the kana-only version has a much smaller vocabulary and a higher repetition rate (higher Rep-N; fourth and fifth columns).

This suggests that correlation dimension captures semantic complexity rather than surface morphological features, detecting semantic repetition rather than morphological repetition.

## D.2 Generating Degenerate Texts

The twenty prompts used in the experiment in Section 5.2 are listed in Table 14.

**Table 13:** Comparison between correlation dimension and Rep-N for two Japanese scripts.

| Book ID | CorrDim | | Rep-N | |
|---|---|---|---|---|
| | normal (kanji + kana) | syllabic (kana only) | normal (kanji + kana) | syllabic (kana only) |
| 31617 | 6.14 | 6.72 | 0.57 | 0.76 |
| 31757 | 6.47 | 6.66 | 0.58 | 0.73 |
| 32941 | 6.44 | 5.76 | 0.66 | 0.84 |
| 32978 | 6.77 | 7.68 | 0.66 | 0.85 |
| 33307 | 6.35 | 6.44 | 0.58 | 0.75 |
| 34013 | 6.36 | 6.60 | 0.57 | 0.75 |
| 34636 | 6.15 | 6.61 | 0.59 | 0.77 |
| 35327 | 6.87 | 6.77 | 0.62 | 0.82 |
| 36459 | 6.53 | 6.28 | 0.65 | 0.82 |
| 37626 | 6.36 | 6.15 | 0.51 | 0.69 |
| Mean | 6.44 | 6.57 | 0.60 | 0.78 |

**Table 14:** List of prompts used in the experiment of degeneration detection.

| No. | Prompt |
|---|---|
| 1 | Describe the primary goals of an effective team. |
| 2 | Explain the basic steps involved in a standard project workflow. |
| 3 | Outline the advantages of using modern technology in daily life. |
| 4 | Discuss the key features of a reliable customer service program. |
| 5 | Summarize the benefits of maintaining a consistent work schedule. |
| 6 | Describe how a typical training session should be conducted. |
| 7 | Explain why clear communication is important in organizations. |
| 8 | Outline the main characteristics of a successful leadership style. |
| 9 | Discuss the factors that contribute to a smooth operational process. |
| 10 | Summarize the core principles of quality assurance. |
| 11 | Describe the role of feedback in performance improvement. |
| 12 | Explain the importance of setting realistic goals. |
| 13 | Outline the steps for conducting a standard evaluation. |
| 14 | Discuss how data is used to inform business decisions. |
| 15 | ummarize the benefits of a structured planning approach. |
| 16 | Describe the essential elements of a professional code of conduct. |
| 17 | "Explain how consistency can improve team efficiency. |
| 18 | Outline the main objectives of a routine maintenance program. |
| 19 | Discuss the value of transparency in reporting results. |
| 20 | Summarize the advantages of applying proven best practices. |

To elicit different types of degenerate texts with GPT-4o, we provided distinct instructions and collected the model's responses as follows.

- **Normal Responses**

You are a creative text generator. Your task is to produce a richly detailed, vivid, and engaging passage of at least 1,500 words (but no more than 2,000) that brings its subject to life with specific imagery, sensory detail, unique examples, and dynamic narrative. Avoid generic descriptions, stock phrases, or "safe" language. Instead, strive for:

- Concrete specifics: name real or invented places, objects, characters, or processes.
- Sensory richness: evoke sight, sound, smell, taste, and touch wherever possible.
- Fresh metaphors and similes: craft original comparisons rather than clichés.
- Varied sentence rhythms: mix short, punchy lines with longer, flowing sentences.
- Emotional or intellectual hooks: give the reader something surprising, thought-provoking, or emotionally resonant.

Generate a single coherent text that feels alive and unmistakably your own.

- **Bland responses**

> You are a text generator. Produce a bland yet coherent passage of 1,500–2,000 words that avoids exact repetition of phrases or words.

- **Incoherent responses**

> You are a text generator. Please produce a richly worded, vivid, and non-repetitive passage (1,500–2,000 words) that is intentionally disjointed—lacking logical coherence—so the reader feels the narrative is fragmented.

- **Repetitive responses** We used the responses generated under **Normal responses** and then extracted the first sentence, and repeated it to form a passage of approximately the same length.

### D.3 Random Text for Stress-Testing LLMs

For stress testing, we used a string of random names separated by commas as the input text. The names are common English names, sampled i.i.d. from a list of unique names. A portion of the resulting text is shown below:

> Quinlan, Anthony, Henry, Felicity, Taylor, Raymond, Xander, Christopher, King, Amanda, Flora, Nicole, Anthony, Frank, Quiana, Owen, Finley, Paige, Victoria, Aaron, Ulrika, Sarah, Ignacio, Emily, Yuna, Imogen, Cameron, Claire, William, Preston, Ulrika, Sabrina, Neil, Zara, Joseph, Orion, Vivian, Quinn, Wyatt, Paul, Sophia, Brian, Flynn, Hayden, Charles, Grace, Carter, Heather, Quest, Jacob, Jordan, Frances, Griffin, Yasmin, Quiana, Penelope, Emma, Sabrina, Elizabeth, Joseph, Zion, Quinlan, Omar, Ruby, Virginia, Ursula, Flynn, Alexander, Ian, Griffin, Frances, Yasmine, Warren, Isaiah, Ryan, Kyle, Xanthe, Lucy, Georgia, Gregory, Ophelia, Georgia, Patricia, Xiomara, Kayla, Finley, Zayden, Noah, Caitlin, Brittany, Connor, Quinton, Urban, King, Blake, Joshua, ...

## E   Consistency with Time-Delayed Embeddings

A potential concern with the correlation dimension estimation method in Section 3 is that we used only the probabilistic information of the next tokens, which is not a complete representation of the model's state. A state of a dynamical system is defined as a point in the phase space, which contains all the information that governs the future evolution of the system.

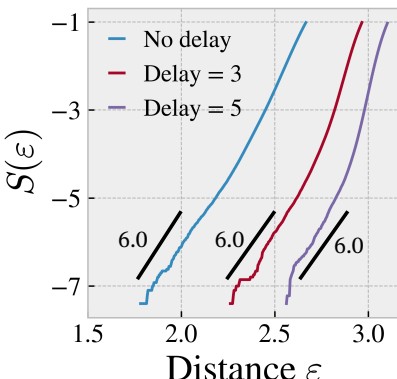

**Figure 8:** Correlation integral curves for time-delayed embeddings with different delays (3 or 5), compared with that for the original sequence.

A common method to reconstruct the phase space from partial observations is to measure the dimension on time-delayed embeddings. While the theoretical effectiveness of time-delayed embedding is

guaranteed by the Takens' theorem and variants [48, 45, 46], the method is empirically sensitive to noise and the choice of the embedding dimension, failing to deliver satisfactory results except for simple, low-dimensional systems. In this work, we aim at characterizing language models that are very high-dimensional and random in nature. For such systems, the time-delayed embeddings are often overwhelmed by observational noise and the dimension is overestimated.

Nevertheless, we observe that the correlation dimension values exhibit good consistency even if the time-delayed embedding is used. For the log-probability vector times series $x = [x_1, x_2, \cdots]$, we acquired two time-delayed sequences $x^{(3)}$ and $x^{(5)}$ with delays of 3 and 5, respectively. For a delay $\tau$, $x_t^{(\tau)} = [x_t; \cdots ; x_{t+\tau-1}]$, i.e., the the concatenation of vectors from $t$ to $t + \tau - 1$. The correlation integral curves for these time-delayed sequences are shown in Figure 8. While the time-delayed embeddings have 3x or 5x the dimension of the original sequence, the correlation integral curves are similar to that of the original sequence at small $\varepsilon$, except that the curves are shifted to the right. The slopes of the curves increased at large $\varepsilon$ because of accumulated noise and is outside the range of interest.

This indicates that the next-token probabilities contain information beyond the next token, and it is sufficient to use the next-token probabilities to characterize the dimension of the model's evolution.

