# OpenReview forum: "Correlation Dimension of Autoregressive Large Language Models"
_NeurIPS.cc/2025/Conference — NeurIPS 2025 poster_

### Official Review · Reviewer_JCEK · 2025-06-30

**Clarity:** 3
**Significance:** 3
**Originality:** 3
**Rating:** 4
**Confidence:** 4

**Summary:**

The authors introduce Correlation Dimension (CD) as a global-structure metric for autoregressive LLMs. Instead of measuring token-level loss, CD treats the log-probability vector sequence
𝑥𝑡 produced during generation as a trajectory in high-dimensional space and estimates its correlation integral
𝑆
(
𝜀
)
∝
$𝜀^𝑑$
 . The slope
𝑑 serves as a proxy for the effective dynamical degrees of freedom: low
𝑑 signals trajectory collapse (e.g., repetition), high
𝑑 indicates random noise, and natural language tends to stabilise around
𝑑
 ⁣
≈
 ⁣
6
 ⁣
−
 ⁣
7. Experiments on Pythia checkpoints, six open LLMs, and GPT-3.5/4-mini demonstrate:

1. a three-phase training curve (short-range learning → long-range capture → compression/overfit);

2. sensitivity to synthetic grammar complexity (Lin-Tegmark);

3. unified detection of repetition, disorder, and semantic sparsity;

4. strong correlation (ρ ≈ 0.95) with HelloEval robustness scores during long-text “name-stream” stress tests.

**Questions:**

1. How do you access the full logits of the API closed model for degradation detection & long text stress experiments? My understanding is that this indicator requires access to the full logits, and the API closed model is difficult to apply?
2. The author shows through the RLHF model that the indicator can compare the same model version across the entire process of "pre-training-instruction fine-tuning-RLHF", but this is not entirely intuitive. Current reasoning models have complex reward rules injected into the reinforcement learning stage and have different generation strategies. I want to know what the actual impact of RL strategy and the so-called "text structure" on the "complexity" of the evaluation of this indicator is? Can you explain why the impact of RL strategy is limited in the experiment?
3. If you randomly permute
𝑥𝑡​ order (destroying long-range structure) while keeping unigram distribution, how does CD react? This would support the “global hierarchy” claim.

**Ethical Concerns:**

["NO or VERY MINOR ethics concerns only"]

**Final Justification:**

This work is well-constructed and interesting. Further exploration of its relationship to the model's generalization and extrapolation capabilities would be even more valuable. Given that its application value has yet to be fully verified, my current rating is objective.

**Limitations:**

Multi-language and code generation have not been tested yet → The transferability of indicators is unknown.

The threshold range needs to be set manually, and there is currently no adaptive algorithm.

Authors should add discussion and mitigation on potential bias against low-resource languages with inherently repetitive morphology (e.g., language-conditioned normalisation).

**Paper Formatting Concerns:**

None.

**Quality:**

3

**Strengths And Weaknesses:**

Strengths
1. The explanation is intuitive. Small distance → The interval [s, t) can be regarded as a "skippable segment", corresponding to text jumps of different scales (skip); the threshold size forms a hierarchy from word level to sentence level, which can cover the global complexity measurement characteristics to a certain extent.
High dimension → The state trajectory occupies a higher dimensional manifold (complex/random); low dimension → The trajectory collapses to a low dimensional attractor (repetitive or pattern-poor). It can monitor phased learning and potential pattern collapse without increasing training overhead, providing signals for early stopping and model selection.
2. The application of long text generation health has certain practical value. Online sliding window calculation d in the reasoning flow: If the continuous window d ↓ < 3 ⇒ trigger rewriting/temperature heating, and warn of "deviant" paragraphs. d also decreases for repetition, disorder, and information sparseness → simpler than multiple dedicated probes.

Weaknesses:
1. When there are lists or rhymed texts, the dimensionality difference between models is large and the explanation is not sufficient.
2. Core estimator (Grassberger–Procaccia) is classical; novelty lies in application, not algorithm. Prior LLM work has used spectrum-based complexity metrics. A side-by-side ablation (same texts) would clarify relative advantages.

---

> ### Author Rebuttal · Authors · 2025-07-30
>
> We greatly appreciate your insightful comments and valuable feedback. We will revise our paper to address all concerns. Below, we respond point-by-point (quoted comments are lightly rephrased due to length limitation).
>
> ---
> > Strength 1. The explanation is intuitive. Small distance → The interval [s, t) can be regarded as a "skippable segment", corresponding to text jumps of
> different scales (skip) ... It can monitor phased learning and potential pattern
> collapse without increasing training overhead, providing signals for early
> stopping and model selection.
>
> Your understanding is exactly right. We’re delighted that our explanation came across intuitively.
>
> ---
> > Strength 2. The application of long text generation health has certain practical value.
> Online sliding window calculation d in the reasoning flow: If the continuous
> window d ↓ < 3 ⇒ trigger rewriting/temperature heating, and warn of "deviant"
> paragraphs. d also decreases for repetition, disorder, and information
> sparseness → simpler than multiple dedicated probes.
>
> We agree that a sliding‐window approach is practical for monitoring correlation dimension in real time. Thank you for this suggestion—we will include it as an explicit use case in the revised manuscript.
>
> ---
> > Weakness 1. When there are lists or rhymed texts, the dimensionality difference between
> models is large and the explanation is not sufficient.
>
> Thank you for highlighting this important issue. We will add a dedicated section discussing it. In brief, we believe the observed difference is a feature rather than a bug. Specifically, the difference reflects a contrast between generalization (low CorrDim) vs. memorization (high CorrDim) behaviors.
>
> In Appendix B, we reported that on a list of scholars’ names, Falcon3-10B yields CorrDim = 8.49, whereas Qwen2.5-7B gives 3.56.
> To investigate the origin of the large difference, we asked the models to complete the name list, as follows:
> - **Input:** “… André Gounelle, Rem B. Edwards, (start generation)”
> - **Gold answer**: Delwin Brown, **David A. Pailin**, **Franklin I. Gamwell**, Forrest Wood, **David Ray Griffin**, …
> - **Falcon3-10B output**: **David A. Pailin**, **Franklin I. Gamwell**, George L. Kline, André Gounelle, Nancy Murphy, **David Ray Griffin**, …
> - **Qwen2.5-7B output**: Thomas J. Altizer, Richard H. Connell, John Basinger, Daniel S. Brown, Robert N. Bellah, …
>
> Falcon3-10B has clearly memorized the list during training, while Qwen2.5-7B treats it as random names and “simplifies” the pattern, hence the lower CorrDim. This case illustrates that correlation dimension not only reflects the text’s structural complexity but also exposes the difference between a model’s memorization and its generalization behaviors.
>
> ---
> > Weakness 2. Core estimator (Grassberger–Procaccia) is classical; novelty lies
> in application, not algorithm. Prior LLM work has used spectrum-based complexity
> metrics. A side-by-side ablation (same texts) would clarify relative advantages.
>
> Thank you for raising this point and suggesting missed references. We will add discussion of spectrum‐based metrics and perform ablations on identical texts to compare their behavior with CorrDim.
>
> ---
> > Question 1. How do you access the full logits of the API closed model for degradation
> detection & long text stress experiments?
>
> There are two ways to apply our method to evaluate a model: (1) running the
> target model itself, or (2) using another model to analyze the text generated by
> the target model. These correspond to Sections 4 and 5 of the paper,
> respectively.
>
> For closed models, approach (1) is unavailable since we don't have access to the
> full logits, but approach (2) remains feasible. For example, in Section 5.1 we
> used the open model Falcon3-10B to evaluate text generated by the closed GPT-4o
> model.
>
> ---
> > Question 2. I want to know what the actual impact of RL strategy and the
> so-called "text structure" on the "complexity" of the evaluation of this
> indicator is? Can you explain why the impact of RL strategy is limited in the
> experiment?
>
> We agree that instruction fine-tuning can affect model behavior.
> We ran a preliminary experiment on the SEP dataset (60 texts) comparing base vs. instruct versions of Qwen2.5 and Falcon3 models:
>
> | Model       | Base CorrDim | Instruct CorrDim | Increase |
> |-------------|----------------|---------------|---------|
> | Falcon3-10B |  6.56          | 6.93          | 0.37 |
> | Falcon3-7B  |  6.56          | 6.81          | 0.25 |
> | Falcon3-3B  |  6.11          | 6.44          | 0.33 |
> | Falcon3-1B  |  6.04          | 6.33          | 0.29 |
> | Qwen2.5-32B |  6.32          | 6.45          | 0.13 |
> | Qwen2.5-14B |  6.26          | 6.33          | 0.07 |
> | Qwen2.5-1.5B |  5.97         | 6.00          | 0.03 |
>
> Instruction fine-tuning consistently raises CorrDim, especially for larger models. We suspect this reflects stylistic shifts from plain text to instruction-driven inputs, but the precise mechanism remains to be explored. We will include these results and discuss possible causes.
>
> ---
> > Question 3.  If you randomly permute $x_t$ order (destroying long-range structure) while
> keeping unigram distribution, how does CD react? This would support the “global
> hierarchy” claim.
>
> After shuffling token order, CorrDim jumps above 10, indicating exactly destroying of long-range structure. We will add this experiment. Thank you for your suggestion.
>
> ---
> > Limitation 1. Multi-language and code generation have not been tested yet → The transferability of indicators is unknown.
>
> We performed quick experiments on other languages and code:
> 1. Natural languages: 80 Project-Gutenberg books (10 books each in English, French, German, Spanish, Italian, Dutch, Chinese, Japanese), processed by Qwen2.5-32B:
>
> | Language | CorrDim (mean)   | CorrDim (median) |
> |---|----------|--------|
> | English  | 6.67    | 6.69  |
> | French   | 6.72    | 6.72     |
> | German   | 7.34       | 7.29           |
> | Spanish  | 6.10    | 6.20           |
> | Italian  | 6.67    | 6.69           |
> | Dutch    | 6.77    | 6.77           |
> | Chinese  | 7.61             | 6.90           |
> | Japanese | 6.98             | 7.05           |
>
> Most languages exhibit a CorrDim of around 6–7. Small variations likely reflect differences in genre and writing systems: our Spanish and English samples are primarily novels; the German corpus consists mostly of scholarly works (e.g., Hegel’s philosophical texts); and the Chinese and Japanese books employ traditional character sets that differ from the modern scripts on which the model was trained.
>
> 2. Code: 30 files each from Python (CPython stdlib), Java (OpenJDK java.base), and C (Linux kernel):
>
> | Language | Mean CorrDim (std) |
> |----------|-------------|
> | Python   | 5.05 (0.73) |
> | Java     | 4.93 (0.89) |
> | C        | 4.94 (0.55) |
>
> Code consistently shows lower CorrDim than natural language, reflecting its higher structural regularity.
> We will integrate these findings and discuss how degeneration phenomena differ between code, natural-language, and complex reasoning tasks.
>
> ---
> > The threshold range needs to be set manually, and there is currently no adaptive algorithm.
>
> We used the simple formula from Appendix A—dependent only on token sequence length and invariant to embedding‐distance scaling—for all experiments. We will clarify this procedure in the main text.
>
> ---
> > Limitation 3. Authors should add discussion and mitigation on potential bias against low-resource languages with inherently repetitive morphology (e.g., language-conditioned normalisation).
>
> Thank you for raising this important point. We agree that careful consideration is needed when evaluating languages with limited resources or highly repetitive morphological structures.
>
> We would like to clarify that correlation dimension (CorrDim) is designed to capture the dynamical properties of the text generation process, rather than surface-level features such as vocabulary size or overt repetition. As such, it is less sensitive to superficial morphological differences and more focused on long-range dependencies embedded in the generation trajectory.
>
> To empirically examine this, we conducted an experiment using three Japanese literary texts, each rendered in two different scripts:
> - Kanji + kana (vocabulary size > 2,000): the standard orthographic system combining ideographic and phonographic characters;
> - Kana only (vocabulary size < 200): a fully phonographic version using syllabary alone.
>
> These versions are parallel in content—that is, they express the exact same meaning and structure, differing only in their script and vocabulary richness.
>
> | Book | Kanji & Kana (vocab > 2000) | Kana only (vocab < 200) |
> |------|----------|--------------|
> | Shisei | 6.72 | 6.65 |
> | Doko e | 7.47 | 7.72 |
> | Kesshouki | 6.51 | 6.62 |
>
> As shown, the correlation dimensions remain stable across these drastically different surface representations. This suggests that CorrDim is indeed robust to changes in vocabulary size and script, and is more reflective of deeper structural properties.
>
> Additionally, as demonstrated in our Lin–Tegmark experiment (Section 4.1), CorrDim varies significantly (from ≈1 to >20) depending on the strength of long-range dependencies, even when applied to simple binary sequences (e.g., 0/1 grammars). This further supports our claim that CorrDim responds primarily to temporal structure, rather than surface repetition.
>
> Even when CorrDim is used to detect degeneration phenomena like repetition, it does so by analyzing the internal trajectory of the LLM’s probability distributions—not by simply counting repeated tokens on the surface, like Rep-N or Self-BLEU scores in previous works.
>
> We will include this analysis and discussion in the revised manuscript, along with a broader reflection on potential biases and promising directions for future investigation.
>
> ---
> We trust these additions and clarifications will address your concerns. Thank you again for your plentiful constructive feedback.

---

> > ### Comment · Reviewer_JCEK · 2025-08-05
> >
> > Thank for the authors' thoughtful response. I was somewhat surprised by the new conclusion: "The correlation dimension not only reflects the structural complexity of the text but also reveals the difference between the model's memorization and generalization behaviors." This seems relevant to "**Generalization or Memorization: Evaluating Data Contamination for Large Language Models**". Based on this work and the definition of CorrDim, I understand quantitatively that this is possible. But qualitatively, these two characteristics appear independent. If the metric can capture both characteristics simultaneously, then how should the weighting be balanced? Is there a unified explanation?
> >
> > I'm not entirely sure whether the external model evaluation method is accurate. Because the weights and parameters are inconsistent, the probability vector sequence also seems inconsistent.
> >
> > Taking all this information into account, overall, this work is well-constructed and interesting. Further exploration of its relationship to the model's generalization and extrapolation capabilities would be even more valuable. Given that its application value has yet to be fully verified, my current rating is objective.

---

> > > ### Author Response · Authors · 2025-08-06
> > > **Thank you for your comments!**
> > >
> > > > This seems relevant to "Generalization or Memorization: Evaluating Data Contamination for Large Language Models".
> > >
> > > We appreciate your comments and the suggested reference. We will look into this direction and add appropriate citations.
> > >
> > > ---------------------------
> > > > I was somewhat surprised by the new conclusion: "The correlation dimension not only reflects the structural complexity of the text but also reveals the difference between the model's memorization and generalization behaviors." I understand quantitatively that this is possible. But qualitatively, these two characteristics appear independent. If the metric can capture both characteristics simultaneously, then how should the weighting be balanced? Is there a unified explanation?
> > >
> > > Yes, these two characteristics are indeed somewhat independent: one pertains to the text, while the other concerns the model. Whether they are independent depends on whether the text under examination is well understood by the model.
> > >
> > > When the text is well understood—such as most articles in the SEP dataset—the model can recognize it with little ambiguity, and the CorrDim reflects the structural complexity of the text. Even so, we do observe small differences in CorrDim between models (Figure 2), which arise from differences in the model’s perception rather than from randomness.
> > >
> > > When the structure of the text is ambiguous to the model—as in zero-shot or in-context learning scenarios—CorrDim may vary significantly between models. One such example is the outlier text “process-theism” that we discussed earlier.
> > >
> > > One advantage of our approach is that we can manipulate the text (as in a benchmark setting) to control the relative influence of these two characteristics—textual complexity and model perception—thereby “tuning” the balance between them.
> > >
> > > To offer a unified view, we propose that CorrDim reflects the **perceived complexity** (line 148, Section 4) of the text as interpreted by the model. We were initially unsure what exactly this “perception” entailed, but thanks to your insightful question, we are now more convinced that it relates to the model’s tendency to generalize or memorize (or contamination).
> > >
> > >
> > > ---------------------------
> > > > I'm not entirely sure whether the external model evaluation method is accurate. Because the weights and parameters are inconsistent, the probability vector sequence also seems inconsistent.
> > >
> > > We apologize for the confusion. The external model evaluation method serves a different purpose from that of using the target model itself; it is not intended as a substitute.
> > >
> > > Using the target model (static aspect): We evaluate the model as-is—that is, we do not ask it to generate any text. Instead, we observe its log-probability vector sequence on human-written text. This corresponds to Sections 3 and 4 of the paper.
> > >
> > > Using an external model (generation aspect): We ask the target model to generate text, and then use an external model to evaluate the generated output. This corresponds to Section 5 of the paper.
> > >
> > > These two approaches are complementary, not interchangeable. Naturally, the external model evaluation is less precise, as we do not have access to the target model’s full logits.
> > >
> > >
> > > > Taking all this information into account, overall, this work is well-constructed and interesting. Further exploration of its relationship to the model's generalization and extrapolation capabilities would be even more valuable. Given that its application value has yet to be fully verified, my current rating is objective.
> > >
> > > Thank you very much for your encouraging comments and constructive feedback.

---

> > > > ### Comment · Reviewer_JCEK · 2025-08-06
> > > >
> > > > Thanks to the authors for their additions and efforts.

---

### Official Review · Reviewer_mbgA · 2025-07-02

**Clarity:** 4
**Significance:** 3
**Originality:** 4
**Rating:** 4
**Confidence:** 4

**Summary:**

While perplexity is widely used as a metric for evaluating LLMs, it has limitations in capturing the long-range structural complexity of natural language, such as repetition when perplexity is low. This paper introduces a novel evaluation metric based on correlation dimension, a measure of self-similarity from fractal geometry, to quantify hierarchical and repetitive structures in language.
The authors demonstrate through experiments that this method can consistently detect various forms of text degeneration, including repetition, incoherence, and blandness, which have traditionally been difficult to quantify.

**Questions:**

Please see the weaknesses listed above.

**Ethical Concerns:**

["NO or VERY MINOR ethics concerns only"]

**Final Justification:**

Based on the rebuttal and the other reviews, particularly in terms of the scope, potential impact, and novelty of the work, I will keep my current scores, leaning slightly toward acceptance. I assume that the authors will address the questions and concerns raised by the reviewers in the final version of the paper.

**Limitations:**

yes

**Quality:**

4

**Strengths And Weaknesses:**

##### Strengths
- This paper proposes a new interesting evaluation metric based on fractal geometry (correlation dimension) that captures long-range structural complexity and generation dynamics of LLMs, which conventional metrics fail to measure.

- Authors demonstrate that the proposed metric can unify and quantify multiple types of text degeneration (e.g., repetition, incoherence, and blandness) that are hard to assess using existing methods.

- The paper is well written and easy to follow.

- The approach is theoretically grounded, taken from established mathematical frameworks such as fractal geometry and dynamical systems theory.

##### Weaknesses
- While perplexity is broadly applicable and robust across different datasets and floating-point precision formats (e.g., FP16), the correlation dimension may yield different optimal values depending on these factors, potentially limiting its general applicability.

- To strengthen the generality and robustness of the findings, it would be helpful to show how the correlation dimension behaves on other text domains (e.g., WikiText-103, BookCorpus), and whether phenomena such as the three-stage evolution are also observed in these settings.

- (minor) The abbreviation "SEP" (Stanford Encyclopedia of Philosophy) is defined only in the appendix. For readability, the full name should also be spelled out in the main text.

---

> ### Author Rebuttal · Authors · 2025-07-31
>
> > This paper proposes a new interesting evaluation metric based on fractal
> geometry (correlation dimension) that captures long-range structural complexity
> and generation dynamics of LLMs, which conventional metrics fail to measure.
> Authors demonstrate that the proposed metric can unify and quantify multiple
> types of text degeneration (e.g., repetition, incoherence, and blandness) that
> are hard to assess using existing methods. The paper is well written and easy to
> follow. The approach is theoretically grounded, taken from established
> mathematical frameworks such as fractal geometry and dynamical systems theory.
>
> Thank you. We are glad you found our work interesting. We appreciate your
> suggestsions and will incorporate them into the revised manuscript.
>
>
> ------------------------------
> > Weakness 1: While perplexity is broadly applicable and robust across different
> datasets and floating-point precision formats (e.g., FP16), the correlation
> dimension may yield different optimal values depending on these factors,
> potentially limiting its general applicability.
>
> Thank you for raising this thoughtful concern. You are absolutely right that correlation dimension reflects both the properties of the text and the model, and therefore may vary accordingly. In fact, this sensitivity is part of its strength---it allows CorrDim to capture meaningful distinctions in structure and generalization behavior across models and datasets.
>
> That said, we agree it is important to assess the method’s robustness with respect to technical factors such as floating-point precision. We sincerely appreciate your suggestion, which prompted us to run additional experiments using Qwen2.5 models quantized to 4 bits with the activation-aware quantization (AWQ) method. We found that the mean correlation dimension remains remarkably stable, with only minimal changes:
>
> | Model | Mean CorrDim (4-bit AWQ) | Mean CorrDim (FP16) |
> |-------|----------------|---------------|
> | Qwen2.5-1.5B-Instruct-AWQ | 6.06  | 6.00  |
> | Qwen2.5-7B-Instruct-AWQ   | 6.44  | 6.34  |
> | Qwen2.5-14B-Instruct-AWQ  | 6.43  | 6.33  |
>
> These results suggest that CorrDim is robust even under aggressive quantization, further supporting its potential for broader applicability. We will include these findings in the revised manuscript, and we are grateful to you for highlighting this important point.
>
>
> > Weakness 2: To strengthen the generality and robustness of the findings, it would be
> helpful to show how the correlation dimension behaves on other text domains
> (e.g., WikiText-103, BookCorpus), and whether phenomena such as the three-stage
> evolution are also observed in these settings.
>
> We agree. We evaluated ten articles from WikiText-103 and ten novels from
> BookCorpus and both datasets showed CorrDim similar to that reported in the paper:
>
> | Dataset | Mean CorrDim (std) |
> |---------|--------------------|
> | WikiText-103 |  6.41 (0.99) |
> | BookCorpus (novels) | 7.03 (1.08) |
>
> In addition, both datasets exhibit the same three-stage training evolution (as in Figure 5a).
> We will include a new figure for these results.
>
> In addition, following Reviewer JCEK’s recommendation, we extended our evaluation to:
> - Seven other natural languages: French, German, Spanish, Italian, Dutch, Chinese, and Japanese
> - Three programming languages: Python, Java, and C
> - Models with and without instruct/RL fine-tuning
>
> The findings are consistent:
> - For natural languages, correlation dimensions consistently range between 6 and 7;
> - For programming languages, they are slightly lower—around 5, reflecting their more rigid structure;
> - Instruction-tuned and RL-fine-tuned models show only modest increases in CorrDim.
>
> We provide a detailed breakdown of these results in our response to Reviewer JCEK (in Limitation 1). We believe these expanded experiments further support the robustness and generality of the correlation dimension as a structural diagnostic across languages, domains, and training paradigms.
>
> ------------------------------
> > Weakness 3: (minor) The abbreviation "SEP" (Stanford Encyclopedia of
> Philosophy) is defined only in the appendix. For readability, the full name
> should also be spelled out in the main text.
>
> Thank you for catching that. We will spell out Stanford Encyclopedia of
> Philosophy (SEP) on first use in the main text.

---

> > ### Comment · Reviewer_mbgA · 2025-08-06
> >
> > Thank you for the clarification. I’m leaning toward accepting the paper, as reflected in my score earlier, and will leave the final decision to the other reviewers and the area chair.

---

### Official Review · Reviewer_fXFE · 2025-07-03

**Clarity:** 2
**Significance:** 4
**Originality:** 4
**Rating:** 5
**Confidence:** 3

**Summary:**

The authors propose a new metric to evaluate LLMs. Their metric is derived from the self-similarity principle and enables the measurement of recursions in natural text. Their experiments demonstrate that this approach handles different types of text effectively, measuring their complexity while also serving as a tool to evaluate imperfections in LLM text distributions.

**Questions:**

See weaknesses

**Ethical Concerns:**

["NO or VERY MINOR ethics concerns only"]

**Final Justification:**

I have read the authors' rebuttal, and they have addressed all my concerns. I remain positive regarding the paper and recommend it for acceptance.

**Limitations:**

yes

**Quality:**

4

**Strengths And Weaknesses:**

Strengths: This is an interesting paper addressing the highly important topic of benchmarking and understanding the quality and structure of LLMs. Its applicability extends beyond autoregressive models (where perplexity is available). The paper addresses many questions that arise during reading through a comprehensive range of experiments.

Weaknesses: The primary question that remains unanswered for me is how to utilize this metric effectively and how to derive meaningful conclusions from it. While this is a specific concern, I believe it reflects how most practitioners who evaluate LLMs approach such tools. For instance, when evaluating Diffusion-based LLMs, there is no convenient method to assess text quality beyond employing numerous metrics (Zipf Coefficient, Self-BLEU, and other metrics depicted in Table 1) that yield disparate results. Consequently, practitioners must consider all these metrics collectively to gain meaningful insights. I have concerns that this metric operates in a similar manner.

More concretely, suppose I have an arbitrary model (for example, my hypothetical Turbo-Scientific-13B model), and I wish to determine whether it outperforms various Llama models of different scales (where the definition of "better" may be arbitrarily complex). Based on the current presentation, it remains unclear to me whether this comparison is feasible. I observe that different models converge toward a specific value of your metric when perplexity is low; however, in an arbitrary case, I lack knowledge of the appropriate convergence value. If my model's metric is higher or lower than the value to which other models converge, what does this indicate? Is it favorable, unfavorable, or indicative of something else entirely?

I acknowledge that you may respond that this question has already been addressed in the paper. However, I would like to emphasize that if this is the case, the answer is not sufficiently prominent. I would appreciate a practical tutorial for practitioners on how to use and interpret this metric, particularly in the absence of extensive pre-existing evaluations (such as knowledge of the convergence value associated with decreasing perplexity across models).

A second, more minor weakness is that it would be valuable to understand how this metric correlates with downstream performance quality. I believe this is ultimately our primary concern, and it would be interesting to determine whether this metric can provide insights into downstream performance.

---

> ### Author Rebuttal · Authors · 2025-07-31
>
> > This is an interesting paper addressing the highly important topic of
> benchmarking and understanding the quality and structure of LLMs. Its
> applicability extends beyond autoregressive models (where perplexity is
> available). The paper addresses many questions that arise during reading through
> a comprehensive range of experiments.
>
> Thank you for your positive feedback. We appreciate your insightful suggestions
> and will incorporate them into the revised version.
>
> --------------------------
> > When evaluating Diffusion-based LLMs, there is no convenient method to assess
> text quality beyond employing numerous metrics (Zipf Coefficient, Self-BLEU, and
> other metrics depicted in Table 1) that yield disparate results.
>
> You are correct. Our method currently applies only to autoregressive LLMs, as it
> relies on next-token probabilities to identify **skip structures** in texts
> (Figure 1a-b). Extending it to diffusion-based models is possible, but it will
> require a method to identify the skip structures from the diffusion process. We
> consider this an important future direction.
>
>
> --------------------------
> > The primary question that remains unanswered for me is how to utilize this
> metric effectively and how to derive meaningful conclusions from it.
> I believe it reflects how most practitioners who evaluate LLMs approach such tools.
>
> Thank you for raising this important point, which was also echoed by other
> reviewers. We completely agree.
>
> In brief, we believe correlation dimension is indicative of a model's tendency
> to **generalize** (low CorrDim) versus **memorize** (high CorrDim). Therefore, a
> model with lower correlation dimension at similar perplexity should be
> considered to generalize better, making CorrDim a potentially useful metric for
> LLM practitioners.
>
> We have conducted an additional case study to support this claim. As shown in
> Table 4 (Appendix B), the correlation dimensions of Qwen2.5-7B (3.56) and
> Falcon3-10B (8.49) showed an exceptionally large difference on the text
> "process-theism" because the two models perceived the long list of scholars'
> names differently. To demonstrate that this high CorrDim in Falcon3-10B is due
> to memorization of the list during pre-training, we asked the two models to
> complete the name list, as follows:
>
> - **Input:** “… André Gounelle, Rem B. Edwards, (start generation)”
> - **Gold answer**: Delwin Brown, **David A. Pailin**, **Franklin I. Gamwell**, Forrest Wood, **David Ray Griffin**, …
> - **Falcon3-10B output**: **David A. Pailin**, **Franklin I. Gamwell**, George L. Kline, André Gounelle, Nancy Murphy, **David Ray Griffin**, …
> - **Qwen2.5-7B output**: Thomas J. Altizer, Richard H. Connell, John Basinger, Daniel S. Brown, Robert N. Bellah, …
>
> As seen, Falcon3-10B memorizes the names in the text (which did not occur in the
> context), while Qwen2.5-7B generalizes to produce generic, plausible names.
>
> This finding echoes with the results in Figure 5a (Section 4.3), where
> correlation dimension decreases in the third training stage as the model learns
> to generalize. In Figure 5b, we also showed how increased correlation dimension
> (near the end of pre-training) coincides with a loss of in-context learning
> accuracy, which requries generalization ability.
>
> Please note that we do not claim that generalization is always better than
> memorization. However, in some challenging tasks, such as complex reasoning,
> generalization (i.e., lower CorrDim) is preferred.
>
>
>
> --------------------------
> > Practitioners must consider all these metrics (Zipf Coefficient, Self-BLEU, and
> other metrics depicted in Table 1) collectively to gain meaningful insights.
> I have concerns that this metric operates in a similar manner.
>
> We agree that correlation dimension, like any single metric, has limitations and
> cannot capture all aspects of LLM behavior. However, as a complexity-inspired
> metric, we believe it offers unique insights---particularly into model
> generalization, as discussed above.
>
> More importantly, correlation dimension is the only metric we are aware of that
> captures the dynamical-system nature of autoregressive LLMs. It is invariant
> across different tokenization schemes or vocabularies---an advantage over other
> metrics such as the Zipf coefficient, Self-BLEU, and Rep-N.
>
> To show this, we conducted a control experiment on the script system in Japanese
> language. We tested three parallel Japanese texts, each rendered in two scripts:
>
> - Kanji + kana: the standard modern mix of ideographic kanji and phonographic kana
> - Kana only: a purely syllabic representation using only kana
>
> “Parallel” here means that both versions convey the exact same meaning,
> differing only in script. The correlation dimensions are as follows:
>
> | Book | Kanji & Kana (vocabulary size > 2000) | Kana only (vocabulary size < 200) |
> |------|----------|--------------|
> | Shisei | 6.72 | 6.65 |
> | Doko e | 7.47 | 7.72 |
> | Kesshouki | 6.51 | 6.62 |
>
> As seen, the correlation dimension remained largely unchanged despite the
> vocabulary size was reduced by an order of magnitude, highlighting correlation
> dimension as a robust metric that captures the underlying dynamical system of
> autoregressive LLMs.
>
> In this case, none of the other metrics (Zipf coefficient, Self-BLEU, Rep-N)
> would work: the Zipf coefficient requires a large vocabulary to manifest,
> Self-BLEU and Rep-N are sensitive to tokenization.
>
>
> --------------------------
> > More concretely, suppose I have an arbitrary model (for example, my hypothetical
> Turbo-Scientific-13B model), and I wish to determine whether it outperforms
> various Llama models of different scales (where the definition of "better" may
> be arbitrarily complex). Based on the current presentation, it remains unclear
> to me whether this comparison is feasible. I observe that different models
> converge toward a specific value of your metric when perplexity is low; however,
> in an arbitrary case, I lack knowledge of the appropriate convergence value. If
> my model's metric is higher or lower than the value to which other models
> converge, what does this indicate? Is it favorable, unfavorable, or indicative
> of something else entirely?
>
> Thank you for your detailed question. As mentioned earlier, we believe a lower
> correlation dimension is indicative of a model's tendency to generalize when
> compared at similar perplexity. In most cases, generalization ability is
> preferred, especially in complex reasoning tasks where the model must generlize
> to unseen contexts.
>
> That said, if your hypothetical Turbo-Scientific-13B model exhibits a lower
> correlation dimension than the Llama models at similar perplexity, it is likely
> to generalize better than the Llama models.
>
> For example, in Figure 2, the Qwen2.5 family (released in 2024) lies to the
> lower right of the Qwen family (2023), Pareto-dominating its predecessor. This
> is expected, as the Qwen2.5 models are trained on more data and employ more
> advanced training techniques.
>
>
>
> > I acknowledge that you may respond that this question has already been addressed
> in the paper. However, I would like to emphasize that if this is the case, the
> answer is not sufficiently prominent. I would appreciate a practical tutorial
> for practitioners on how to use and interpret this metric, particularly in the
> absence of extensive pre-existing evaluations (such as knowledge of the
> convergence value associated with decreasing perplexity across models).
>
> We will add a dedicated section in the revised manuscript to clarify how
> correlation dimension can be used in practice for LLM evaluation. We
> appreciate you pointing out that oversight.
>
>
> --------------------------
> > A second, more minor weakness is that it would be valuable to understand how
> this metric correlates with downstream performance quality. I believe this is
> ultimately our primary concern, and it would be interesting to determine whether
> this metric can provide insights into downstream performance.
>
> We agree that the ultimate goal of LLM evaluation is to understand and predict
> how well a model performs on downstream tasks. However, this is extremely
> challenging for LLMs, and most existing LLM evaluations are based on
> fine-grained benchmarks.
>
> To reach a definitive conclusion, we believe it would better to establish a more
> direct theoretical link and CorrDim and the LLM generation process---an
> important direction that lies beyond the scope of this paper. We are interested
> in exploring this in future work and hope that our current findings provide a
> foundation for new research on LLM generalization from a complex-system
> perspective.

---

> > ### Comment · Reviewer_fXFE · 2025-08-04
> >
> > Thank you for your response. It has addressed my concerns. After careful consideration, I have decided that my score remains sufficiently positive, and I am voting for acceptance. Therefore, I have chosen not to increase it further to ensure a fair review.

---

### Official Review · Reviewer_Bw4t · 2025-07-03

**Clarity:** 3
**Significance:** 3
**Originality:** 3
**Rating:** 4
**Confidence:** 4

**Summary:**

The paper introduces a novel framework for quantifying the implicit textual complexity by leveraging “correlation dimension”, a concept from fractal geometry. In particular, the authors adapt the Grassberger–Procaccia algorithm [1] (originally formulated for Euclidean spaces) to the statistical manifold of probability distributions.  The core idea is to treat the sequence of next-word probability distributions (as produced by an autoregressive LLM) as a trajectory in this manifold, and to measure its fractal (self-similar) structure. Empirically, the authors compute the correlation dimension of language data using LLMs [2, 3 ,4].  They find a universal fractal dimension of roughly 6–7 for diverse natural languages (e.g., English, Chinese, Japanese, German) and genres, indicating pervasive global self-similarity. The finding of a common “language dimension” and the comparison to theoretical random processes are particularly noteworthy.

*References*

[1] *Characterization of strange attractors.* Grassberger et al. In Physical Review Letters 1983.

[2] *Pythia: A suite for analyzing large language models across training and scaling.* Biderman et al. In ICML 2023.

[3] *The llama 3 herd of models.* Grattafiori et al. arXiv 2024.

[4] *Qwen2.5 technical report.* Yang et al. arXiv 2024.

**Questions:**

1. The reviewer is curious about how would systematically disrupting long-range correlations (beyond simple word shuffling) affect the dimension?
2. Would the correlation dimension framework extend naturally to structured representations (e.g., syntactic tree), and if so, do the authors foresee the dimension increasing or decreasing in that case?
3. Could the authors relate the correlation dimension to other measures of linguistic complexity? For example, measures like entropy rate, mutual information scaling, or classical fractal exponents (Hurst exponent) have been used for text.

**Ethical Concerns:**

["NO or VERY MINOR ethics concerns only"]

**Final Justification:**

I have updated my score to reflect my decision, considering the sufficient rebuttal the authors have provided.

**Limitations:**

yes

**Paper Formatting Concerns:**

no major formatting issues

**Quality:**

3

**Strengths And Weaknesses:**

**Strengths**:

1. The theoretical demonstration is well-grounded. Extending the Grassberger–Procaccia algorithm to probability distributions is carefully justified.
2. This paper is generally well-written and easy-to-follow.
3. The authors explore on a novel perspective of language modeling by linking it to dynamical systems theory and fractal geometry.

**Weaknesses**:
1. The paper could more explicitly contrast with any related fractal or information-theoretic analyses of text (e.g., multifractal analyses, entropy measures) to highlight what is unique.

2. The practical usages of proposed method are somewhat speculative at this stage. The paper reveals an interesting phenomenon (i.e., language self-similarity) but stops short of demonstrating clear downstream applications. For example, it is not yet shown how knowing this dimension improves language modeling or analysis tasks.

---

> ### Author Rebuttal · Authors · 2025-07-31
>
> We sincerely thank you for taking the time to provide constructive feedback. We
> notice that some of your remarks appear to reflect findings or content from a
> different work, rather than the current one. To avoid any potential confusion,
> let us briefly summarize the unique contributions of this manuscript:
>
> 1. We show that the correlation dimension of text arises from multiscale "skip
> structures" in the language generation process.
>
> 2. We validate that the proposed method remains robust when applied to
> under-trained models and we uncover multiple distinct phases of LLM pretraining
> (Section 4).
>
> 3. We demonstrated that correlation dimension serves as a comprehensive
> indicator of degeneration in long-text generation, capturing repetition,
> incoherence, and blandness under a single metric (Section 5).
>
> We hope this brief summary helps to clarify the focus of this submission. Below,
> we respond to each of your comments in detail.
>
>
> --------------------------------------------
> > Strength: The theoretical demonstration is well-grounded. Extending the
> Grassberger–Procaccia algorithm to probability distributions is carefully
> justified. This paper is generally well-written and easy-to-follow. The authors
> explore on a novel perspective of language modeling by linking it to dynamical
> systems theory and fractal geometry.
>
> Thank you for your encouraging feedback. We are pleased to hear that you found
> our theoretical foundation solid and the writing clear. Your recognition of the
> novel connection between fractal geometry and language modeling is greatly
> appreciated.
>
>
> --------------------------------------------
> > Weakness 1: The paper could more explicitly contrast with any related fractal or
> information-theoretic analyses of text (e.g., multifractal analyses, entropy
> measures) to highlight what is unique.
>
> We agree. In the revised manuscript, we will expand our discussion to include
> comparisons with existing complexity measures, such as multifractal spectra,
> entropy rates, and spectrum-based metrics (as also noted by another reviewer).
>
>
> --------------------------------------------
> > Weakness 2: The practical usages of proposed method are somewhat speculative
> at this stage. The paper reveals an interesting phenomenon (i.e., language
> self-similarity) but stops short of demonstrating clear downstream applications.
> For example, it is not yet shown how knowing this dimension improves language
> modeling or analysis tasks.
>
> We agree that demonstrating downstream relevance is important. We believe our
> correlation dimension (CorrDim) provides a complementary metric for evaluating
> generalization in LLMs, especially when used alongside perplexity.
>
> As illustrated in Figure 2, models in the lower-right corner (low perplexity,
> low CorrDim) tend to generalize better. Furthermore, in Figure 5, we observe a
> decreasing CorrDim trend during the final phase of pretraining, suggesting that
> CorrDim reflects increasing generaliation over time.
>
> To further support this interpretation, we conduted a case study on the outlier
> text *process-theism* (Table 4, Appendix B). There, CorrDim differed
> significantly between Qwen2.5-7B (3.56) and Falcon3-10B (8.49). Upon analysis,
> we found that the higher CorrDim in Falcon3-10B was due to memorization of
> training content, whereas Qwen2.5-7B exhibited better generalization. This
> suggests CorrDim may serve as a useful diagnostic tool for memorization vs.
> generalization, which is a key concern in LLMs.
> (For full details, please refer to our response to Reviewer JCEK in the "Weakness 1" section)
>
>
> --------------------------------------------
> > Question 1: The reviewer is curious about how would systematically disrupting
> long-range correlations (beyond simple word shuffling) affect the dimension?
>
> Disrupting long-range dependencies tends to increase the correlation dimension,
> since CorrDim is sensitive to the global recurrence (or skip structure) in the
> generation trajectory.
> - When the sequence is fully shuffled by word, all long-range correlations are
> destroyed, and the CorrDim typically rises above 10, indicating random-like,
> high-dimensional behavior.
> - In cases of partial disruption (e.g., shuffling paragraphs), long-range structure
> is weakend but not erased. Here, CorrDim increases, but remains below 10.
>
> This trend is demonstrated in our Lin-Tegmark grammar experiments (Figure 3b),
> where we gradually increase the dependency parameter $q$ from 0.001 to
> 0.5---reducing long-range dependencies---the CorrDim rises from below 1 to above
> 10.
>
>
> --------------------------------------------
> > Question 2. Would the correlation dimension framework extend naturally to
> structured representations (e.g., syntactic tree), and if so, do the authors
> foresee the dimension increasing or decreasing in that case?
>
>
> Yes, we believe the framework can be extended to structured representations such
> as syntactic trees, with some caveats. CorrDim is fundamentally driven by
> multiscale recurrence patterns, which can be viewed as analogous to removing or
> skipping subtrees in a hierarchical representation of text.
>
> If a syntactic representation is designed to preserve these skip structures,
> we would expect the resulting CorrDim to be comparable. However, common
> syntactic formalisms (e.g., Universal Dependencies) may not fully capture
> such generative recurrences. In such cases, the measured CorrDim may not
> align with that of the original sequence.
>
>
> --------------------------------------------
> > Question3. Could the authors relate the correlation dimension to other
> measures of linguistic complexity? For example, measures like entropy rate,
> mutual information scaling, or classical fractal exponents (Hurst exponent) have
> been used for text.
>
> Thank you for this insightful question. We will add a dedicated section in the
> revised manuscript to clarify how CorrDim relates to these measures. Here is a
> brief summary.
>
> - Entropy rate (closely related to perplexity): CorrDim tends to converge to
> around 6.5 when entropy rate is low---typically in well-trained LLMs. However,
> when entropy is high (e.g., in early training), CorrDim and entropy rate diverge
> and capture different aspects of the model.
>
> - Mutual information (MI) scaling: Higher CorrDim is generally associated with
> faster decay, as we demonstrated in the Lin-Tegmark experiments (Figure 3).
> However, MI and CorrDim capture complementary facets:
>   - Fixed points, limit cycles, and quasi-periodic tori all exhibit zero MI decay,
>   yet their CorrDim values differ: 0, 1, and 2, respectively.
>   - Thus, CorrDim reflects geometric complexity, while MI scaling reflects predictability.
>
> - Hurst exponent: Hurst exponent measures linear autocorrelation in 1D
> sequences. It aligns with CorrDim only when temporal dependencies are linear
> (e.g., fractional Brownian motions). For nonlinear and high-dimensional dynamics
> (like language generation), CorrDim is generally more sensitive and informative,
> capturing nonlinear dependencies that Hurst exponent may miss.

---

> > ### Comment · Reviewer_Bw4t · 2025-08-04
> >
> > Thanks for the rebuttal. I have carefully read the responses and I believe my concerns have been properly addressed.

---

### Decision · Program_Chairs · 2025-09-17

**Decision:**

Accept (poster)

**Comment:**

This work establishes a measure of text complexity conditioned on a language model based on the self-similarity of the fractal structure of its next-word probability distributions. It establishes a common dimension shared by a wide range of human languages as well as more and less predictable forms of text, and introduces a range of other observations.

Reviewers generally commended this work for its novel and timely insights, grounded approach, and good writing. Reviewers mostly found the metric (itself not a contribution of this work) to be intuitive and informative in this context, and noted that it may have some practical applications, though the extent of those may be modest.

Given this, we recommend accepting the paper. Please update the work as discussed with the reviewers.